# Whole genome CRISPRi screening identifies druggable vulnerabilities in an isoniazid resistant strain of *Mycobacterium tuberculosis*

XinYue Wang [1], William J. Jowsey [1], Chen-Yi Cheung[1], Caitlan J. Smart[1], Hannah R. Klaus [1,2], Noon EJ Seeto[1], Natalie JE Waller[1], Michael T. Chrisp[1], Amanda L. Peterson[3], Boatema Ofori-Anyinam [4,5], Emily Strong[6], Brunda Nijagal[3], Nicholas P. West [6], Jason H. Yang [4,5], Peter C. Fineran [1,2,7,8], Gregory M. Cook [1,2,9], Simon A. Jackson [1,2] & Matthew B. McNeil [1,2] ✉

Drug-resistant strains of *Mycobacterium tuberculosis* are a major global health problem. Resistance to the front-line antibiotic isoniazid is often associated with mutations in the *katG*-encoded bifunctional catalase-peroxidase. We hypothesise that perturbed KatG activity would generate collateral vulnerabilities in isoniazid-resistant *katG* mutants, providing potential pathway targets to combat isoniazid resistance. Whole genome CRISPRi screens, transcriptomics, and metabolomics were used to generate a genome-wide map of cellular vulnerabilities in an isoniazid-resistant *katG* mutant strain of *M. tuberculosis*. Here, we show that metabolic and transcriptional remodelling compensates for the loss of KatG but in doing so generates vulnerabilities in respiration, ribosome biogenesis, and nucleotide and amino acid metabolism. Importantly, these vulnerabilities are more sensitive to inhibition in an isoniazid-resistant *katG* mutant and translated to clinical isolates. This work highlights how changes in the physiology of drug-resistant strains generates druggable vulnerabilities that can be exploited to improve clinical outcomes.

*Mycobacterium tuberculosis*, the primary causative agent of Tuberculosis (TB), remains a significant global health problem. Whilst drug-susceptible (DS) strains of *M. tuberculosis* can be treated with a four-drug four-month regimen (i.e., rifapentine, isoniazid, moxifloxacin and pyrazinamide), drug-resistant (DR) strains necessitate prolonged treatments with reduced cure rates[1,2]. There is a clear need for novel antibiotics and drug combinations that can rapidly sterilise DR-strains and curb the emergence of drug resistance.

Mycobacterial KatG is a bifunctional catalase-peroxidase involved in detoxifying reactive oxygen species (ROS)[3–5]. Although not required for mycobacterial growth in vitro, KatG is required in response to increases in ROS and survival within murine infection models[6,7].

[1]Department of Microbiology and Immunology, University of Otago, Dunedin, New Zealand. [2]Maurice Wilkins Centre for Molecular Biodiscovery, University of Auckland, Auckland, New Zealand. [3]Metabolomics Australia, Bio21 Institute, The University of Melbourne, Melbourne, VIC, Australia. [4]Center for Emerging and Re-emerging Pathogens, Public Health Research Institute, Rutgers New Jersey Medical School, Newark, NJ, USA. [5]Department of Microbiology, Biochemistry and Molecular Genetics, Rutgers New Jersey Medical School, Newark, NJ, USA. [6]School of Chemistry and Molecular Biosciences, The University of Queensland, Brisbane, QLD, Australia. [7]Genetics Otago, University of Otago, Dunedin, New Zealand. [8]Bio-Protection Research Centre, University of Otago, Dunedin, New Zealand. [9]School of Biomedical Sciences, Queensland University of Technology, Translational Research Institute, Woolloongabba, QLD, Australia. ✉e-mail: matthew.mcneil@otago.ac.nz

Isoniazid (INH), a key component of front-line anti-tubercular therapy, requires KatG to form the active INH-NAD adduct that binds to the target InhA[8,9]. Structural changes in KatG are the primary cause of INH-resistance (INH[R]), with >70% of INH[R] clinical isolates having a mutation in *katG*[10]. Whilst a large variety of *katG* mutations have been reported, the majority of clinical INH[R] strains harbour a substitution at KatG Serine 315 to threonine (77.8%), with alternative mutations to asparagine (1%) or arginine (0.1%) being observed at a lower frequency[11].

To mitigate fitness costs associated with drug resistance, specific cellular pathways become more important for the growth of DR-strains, making them more vulnerable to inhibition compared to a DS-parent[12–16]. Prior efforts to identify these collateral vulnerabilities involved systematic antibiotic screening, revealing antibiotics with increased activity against DR-pathogens[17–23]. However, these studies were limited to antibiotics that inhibited only a small number of druggable targets. High-throughput approaches are needed to comprehensively define collateral vulnerabilities at a genome-wide level and to identify novel drug targets to prioritise in future drug development programmes.

CRISPR interference (CRISPRi) uses guide RNA (gRNA) sequences and nuclease deficient Cas9 (dCas9) to transcriptionally silence target genes[24]. Whole genome CRISPRi (WG-CRISPRi) provides a quantifiable measure of fitness cost or vulnerability associated with target inhibition at a genome-wide level and allows for the assessment of both non-essential and essential genes[25–29].

In this work, by combining WG-CRISPRi screens with transcriptional and metabolomic analysis, we have generated a genome-wide map of vulnerabilities in an INH-resistant *katG* mutant of *M. tuberculosis*. This work expands upon recent WG-CRISPRi studies in a rifampicin resistance mutant of *M. tuberculosis* and provides fundamental insights into how changes in the physiology of DR strains generate druggable vulnerabilities that can be exploited to improve clinical outcomes[30].

## Results

### Whole-genome CRISPRi screening to identify genetic vulnerabilities in *M. tuberculosis*

We hypothesised that WG-CRISPRi could be used to identify genes of increased vulnerability in an INH-resistant *katG* mutant of *M. tuberculosis* (KatG[L458QfsX27], hereafter referred to as INH[R]-*katG*). To investigate this, we constructed a WG-CRISPRi plasmid library to transcriptionally repress approximately 96% of annotated genes in *M. tuberculosis* (Fig. 1a and Supplementary data 1). The CRISPRi plasmid pLJR965 encodes a *Streptococcus thermophilus* dCas9 mutant (Sth1dCas9) and a gRNA-scaffold sequence, both of which are induced by anhydrotetracycline (ATc)[24,31,32]. In total, 22,996 gRNAs were selected to target 3991 unique protein-coding genes (Supplementary Data 1). Most genes were targeted by 6 individual gRNAs with variation in the predicted PAM strength, gRNA length and predicted gRNA strength (Supplementary Fig. 1a–d)[24,26]. The WG-CRISPRi library screen was performed in the DS-parent *M. tuberculosis* strain mc²6206 (ΔpanCD, ΔleuCD) by culturing pooled populations for a total of 14 days, with back-dilutions into fresh media with ATc on days 5 and 10 (Fig. 1a). We controlled the level of CRISPRi knockdown by adding different concentrations of ATc: high, medium, or low, using 300, 30, or 3 ng/ml ATc, respectively. At these concentrations of ATc, the level of transcriptional repression was comparable between the DS-parent and INH[R]-*katG* when tested by qPCR (Supplementary Fig. 1e–h). Genomic DNA was harvested on days 5, 10 and 14 and used to generate amplicon libraries for deep sequencing. Amplicon sequencing results demonstrated the number of gRNAs that were depleted by >1-$\log_2$ reduction (i.e., two-fold reduction) in abundance relative to the uninduced (ATc-0) sample increased over time. 2969 gRNAs were depleted on at least one of the sampled time points or ATc concentrations (Fig. 1b, and Supplementary Fig. 1i). Next, we classified genes as "essential" if ≥2 of their targeting gRNAs had >1-$\log_2$ reduction in abundance in the day 14 + ATc-300 sampling condition relative to the uninduced sample. For the DS-

parent mc²6206, 540 genes met our essentiality criteria (Fig. 1c). Consistent with prior work showing variations in the time taken to observe a fitness cost when using CRISPRi, the number of genes that had ≥2 gRNAs with >1-$\log_2$ reduction increased over time (Fig. 1c)[29].

Comparison of our WG-CRISPRi screen to a prior *M. tuberculosis* WG-CRISPRi screen (hereafter referred to as CRISPRi-Bosch) revealed 93.3% agreement in gene essentiality calls, with discrepant essentiality calls for 212 essential and 54 non-essential genes (Supplementary Fig. 1j and Supplementary Data 2)[26]. Of these, most of the 212 genes with discrepant essentiality predictions were defined as low vulnerability in CRISPRi-Bosch (i.e., a vulnerability index (VI) > −5) (Supplementary Fig. 1k–l)[26]. Experimental inspection of genes with discrepant essentiality predictions that we called non-essential, but Bosch called essential (e.g., *blaR* and *menE*) demonstrated that they failed to impair the growth of *M. tuberculosis* when grown under conditions comparable to our WG-CRISPRi screen (Supplementary Fig. 1m). The *menE* targeting gRNA did eventually impair growth, but only over a prolonged time course (Supplementary Fig. 1i). Differences in experimental design, including growth media (e.g., OADC vs OADC + Glycerol in CRISPRi-Bosch), the magnitude of gRNA depletion and the timing of essentiality calls (Day 14 vs Day 24 in CRISPRi-Bosch) are likely contributors to the differences in gene essentiality calls. In conclusion, our WG-CRISPRi platform for *M. tuberculosis* can reproducibly identify relative differences in the fitness costs associated with targeted gene inhibition.

### WG-CRISPRi identifies genes with increased vulnerability in INH[R]-*katG*

We hypothesised that gRNAs targeting pathways that are more vulnerable to inhibition in INH[R]-*katG* would have greater effects on growth in a pooled screen and manifest as either (i) increased depletion in INH[R]-*katG* at day 14 and/or (ii) be depleted from INH[R]-*katG* earlier relative to the DS-parent in the WG-CRISPRi screen (Fig. 1a). WG-CRISPRi screening identified 3820 depleted gRNAs and 631 essential genes in INH[R]-*katG* (Fig. 1d, e and Supplementary Fig. 2a). We defined genes as more vulnerable to inhibition in INH[R]-*katG* when (i) they were essential for growth in INH[R]-*katG* and (ii) had ≥2 gRNAs that were depleted by >1-$\log_2$ fold more than observed in the DS-parent (Fig. 1f–g and Supplementary Fig. 2b). Of the 631 essential genes, we defined 388 genes as "more vulnerable" in INH[R]-*katG* and further classified 168 genes as "synthetic lethal" as they were "essential" in INH[R]-*katG* but "non-essential" in the DS-parent (Fig. 1g). This number of "more vulnerable" genes is in-line with other studies investigating interactions with genes that have core biological functions[33,34].

We hypothesised that gRNAs targeting highly vulnerable essential genes would be maximally depleted at day 14 in the DS-parent and be unable to discern differences in vulnerability. Consistent with this, we identified 38 genes (e.g., *rv0123*, *rplC*, and *rplW*) that met our criteria of being more vulnerable in INH[R]-*katG* at ≥3 sampling conditions (i.e., earlier time points/lower ATc concentrations) but not at day 14 with ATc-300 (Supplementary Fig. 2c, d). Furthermore, RNA sequencing of the DS-parent and INH[R]-*katG* that were not subject to CRISPRi demonstrated there was no correlation between differential gene expression and altered vulnerability, with only three genes of increased vulnerability being differentially expressed in INH[R]-*katG* (Supplementary Fig. 2e and Supplementary data 3). In conclusion, WG-CRISPRi essentiality screens can identify genes that are of increased vulnerability to inhibition in INH[R]-*katG* that cannot otherwise be predicted from changes in gene expression.

### Diverse pathways are more vulnerable to inhibition in INH[R]-*katG*

Having identified genes that were more vulnerable to inhibition at a genome-wide level, we sought to identify pathways that were enriched for genes of increased vulnerability in INH[R]-*katG*. Pathway analysis demonstrated that nucleotide metabolism, protein synthesis, respiration and amino acid metabolism were the most vulnerable functional

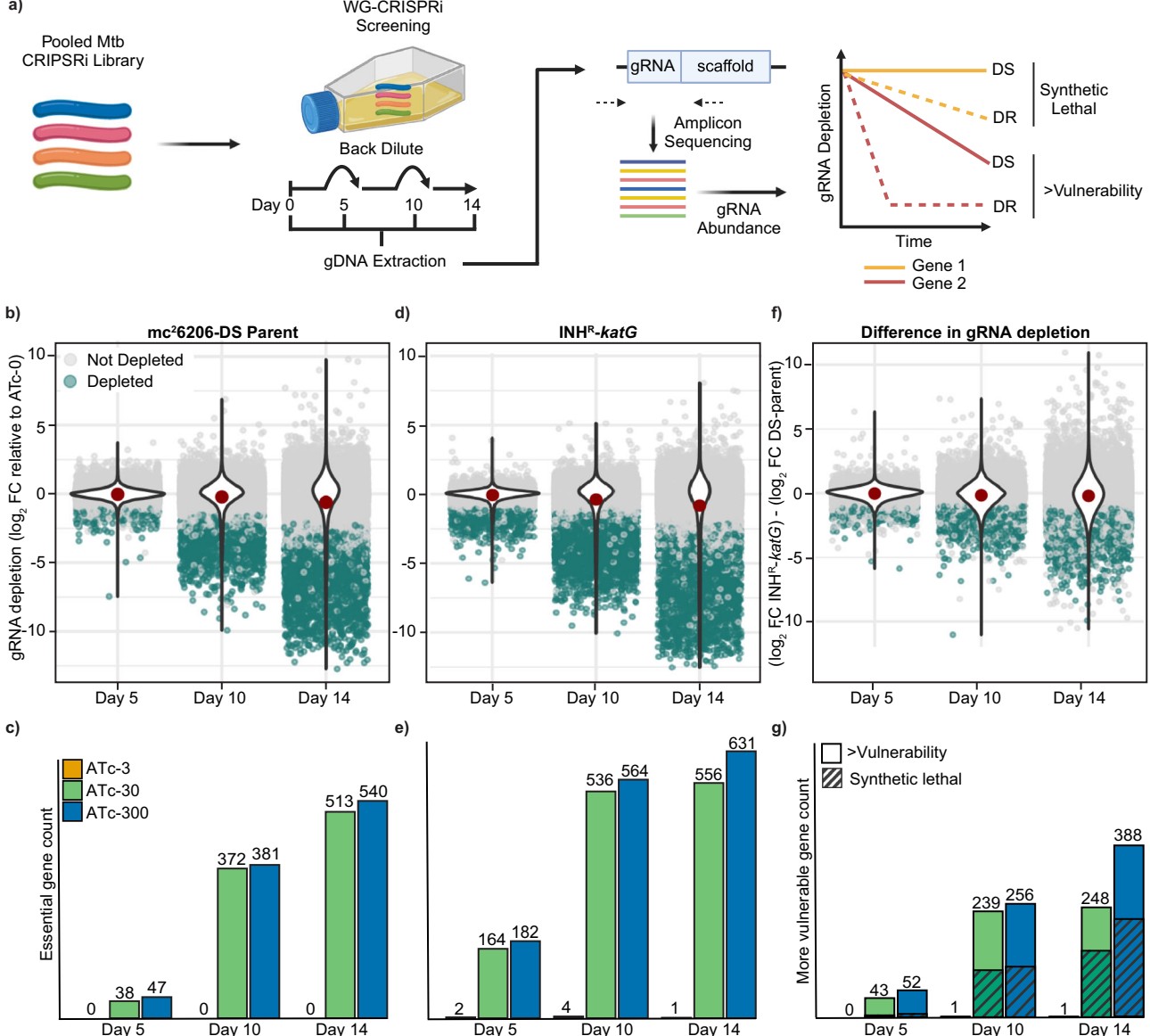

**Fig. 1 | WG-CRISPRi screening in *M. tuberculosis* strain mc26206. a** Experimental design for WG-CRISPRi essentiality screening. The pooled CRISPRi plasmid library is transformed into appropriate *M. tuberculosis* strains. WG-CRISPRi screens were performed for 14 days, with varying concentrations of ATc and at day 5 and 10 back diluted (1/20) into fresh media + ATc to maintain log phase growth. gDNA was extracted, gRNA amplified and sequenced from samples collected on days 5, 10 and 14. The proportion of each gRNA at each ATc concentration within the pooled population was quantified relative to ATc-0 for each sampled time point and plotted on a log2-fold scale as a reduction in gRNA abundance. Increased depletion of gRNA was used to identify genes that are either synthetic lethal or have increased vulnerability in INH^R-*katG*. Created in BioRender. Wang, X. (2023) BioRender.com/q45z115. **b**–**g** Summary of gRNA abundance in the *M. tuberculosis* strain (**b**) mc²6206 DS-parent, (**d**) INH^R-*katG*, and (**f**) the depletion difference between the DS-parent and INH^R-*katG*. The gRNA abundance is relative to the ATc-0 control at each

time point calculated by the exact test and *p*-values adjusted by the Benjamini–Hochberg method. Unchanged gRNAs are coloured grey, whilst gRNA with significant > 2-fold depletion (Benjamini–Hochberg adjusted *p* < 0.01) are coloured green. The red dot within each violin plot denotes the mean gRNA depletion. A total of 3991 unique protein-encoding genes were screened. Of these, the number of essential genes identified in the (**c**) DS-parent and (**e**) INH^R-*katG* were illustrated under different time points and ATc concentrations. **g** Genes identified as being more vulnerable to inhibition in INH^R-*katG* are defined as when (i) they were called essential in INH^R-*katG* and (ii) had no less than 2 gRNAs that were depleted by >1-log2 fold more than observed in the DS-parent. Of these, genes were identified as "synthetic lethal" when had no less than 2 gRNAs that were (1) not depleted in the DS-parent and (2) were depleted by >1-log2 fold in INH^R-*katG* more than observed in the DS-parent. Source data are provided as a Source Data file.

classes (Fig. 2a and Supplementary Data 4). Whilst, cell envelope and DNA processing classes were not highly affected, >20% of genes in the cell wall synthesis and DNA replication subclasses were more vulnerable to inhibition (Supplementary Figs. 2f, 3 and Supplementary Data 4). To validate the accuracy of our screen, we selected 30 gRNAs targeting genes from diverse functional classes for follow-up testing. We hypothesised that gRNAs targeting these genes would require less CRISPRi repression (i.e., ATc) to impair the growth of INH^R-*katG*

compared to the DS-parent. In ATc dose response assays, 18 gRNAs, including gRNAs targeting *gyrB*, *atpD*, *atpF*, *rpoB*, *hadA*, *kasA* and *mmpL3*, required less ATc to inhibit the growth of INH^R-*katG* compared to the DS-parent (Fig. 2b–d and Supplementary Fig. 3a–d). gRNAs targeting *clpC1* and *iscS* did not reduce the ATc MIC but altered the shape of the dose response curve (Fig. 2e and Supplementary Fig. 3e). Most gRNAs that required less CRISPRi repression to impair growth also had improved killing against INH^R-*katG* (Fig. 2f and Supplementary

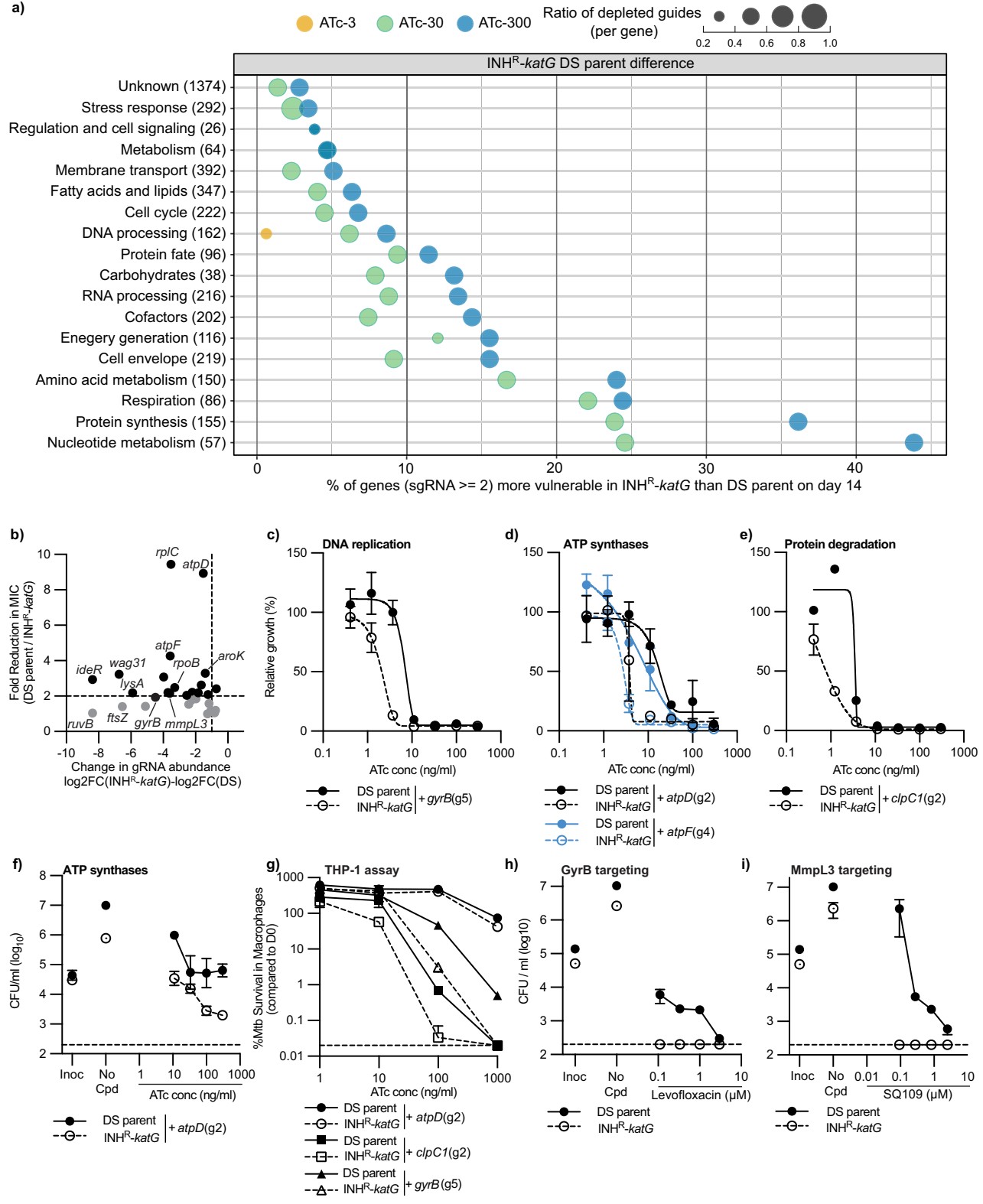

Fig. 3f–m). Although the magnitude of improved killing varied, gRNAs targeting *atpD* and *kasA* were static against the DS-parent yet killed INH[R]-*katG* (Fig. 2f and Supplementary Fig. 3f). We hypothesised that differences in experimental setup were responsible for some gRNAs showing a poor correlation between the level of gRNA depletion in our screen and fold difference in ATc MIC[29]. Specifically, some genes when targeted by CRISPRi have a buffering phenotype where multiple cell divisions are needed to reduce a target protein below a critical

threshold to observe a phenotype. For this reason, some genes do not have a strong phenotype in ATc dose-response assays where the cultures are not back-diluted, whilst they have a strong phenotype in the genetic screen where they are back-diluted.

We hypothesised that if the increased vulnerability of target genes translated to a host-relevant model, then gRNAs would have a greater effect on the intracellular survival of INH[R]-*katG* within a THP-1 macrophage infection model. Consistent with this, gRNAs targeting *gyrB*,

**Fig. 2 | Pathway analysis identifies diverse functions that are more vulnerable to inhibition in INH$^R$-katG. a** Pathway analysis identifies functional classifications of genes with increased vulnerability in INH$^R$-katG (i.e., Fig. 1g). The bubble plot represents data from day 14. The x-axis quantifies the proportion of genes called "more vulnerable" over the total number of *M. tuberculosis* genes per functional pathway, and the y-axis shows the name of each functional pathway with the total number of genes in each class labelled in brackets. Within each functional class, the dot size indicates the average ratio of gRNAs targeting each gene that is more depleted in INH$^R$-katG. The dot colour denotes the ATc concentration from which the amplicon sequencing was performed. **b** Scatter plot illustrating experimental validation of each of the 30 gRNAs. Data is plotted as the difference in gRNA abundance between INH$^R$-katG and the DS-parent from WG-CRISPRi screens (Day14+ATc-300) (x-axis) against the fold change in ATc MIC between INH$^R$-katG and the DS-parent from ATc dose response assays (y-axis). The vertical dotted line indicates the cut-off for a gRNA to be significantly depleted, whilst the horizontal dotted indicates a two-fold change in ATc MIC. **c**–**e** Growth of *M. tuberculosis* DS- parent and INH$^R$-katG expressing for gRNA targeting (**c**) *gyrB*, (**d**) *atpD* and *atpF* and (**e**) *clpC1* in ATc dose response assays (mean ± extrema of two biological replicates, *n*=3 independent experiments). The (gx) after each gRNA denotes the specific gRNA targeting each gene. **f** Viability plots of *M. tuberculosis* DS-parent and INH$^R$- katG expressing for gRNA targeting *atpD* (mean ± extrema of two biological replicates, n=3 independent experiments). **g** *M. tuberculosis* survival in macro- phages. THP-1 macrophage cells were infected with *M. tuberculosis* DS-parent and INH$^R$-katG cells expressing the stated gRNA (mean ± SD of three biological repli- cates, *n* = 2 independent experiments). **h, i** The susceptibility of *M. tuberculosis* DS- parent and INH$^R$-katG to increasing concentrations of (**h**) levofloxacin (mean ± extrema of two biological replicates, n=3 independent experiments) and (**i**) SQ109 (mean ± extrema of two biological replicates, *n* = 3 independent experiments). **f, h, i** Inoc denotes the starting CFU/ml and no-cpd denotes the detected CFU/ml in the absence of compound Dashed line represents the lower detection limit. Source data are provided as a Source Data file.

*clpC1* or *rpoB* caused a greater reduction in viable colonies of INH$^R$- *katG* compared to the DS-parent (Fig. 2g and Supplementary Fig. 3n)[35]. The gRNA targeting *atpD* had no effect on the intracellular survival of INH$^R$-*katG* (Fig. 2g). We next hypothesised that if genes of increased vulnerability could serve as druggable vulnerabilities, then INH$^R$-*katG* would have increased sensitivity to killing by antibiotics that targeted these pathways. Whilst INH$^R$-*katG* was not more sensitive to growth inhibition, it had a large increase in sensitivity to killing by levoflox- acin (GyrB targeting), SQ109 and CPD1 (MmpL3 targeting), bortezomib (putative Clp targeting), thiocarlide (HadAB targeting) and, bedaqui- line (BDQ) and BB16F (ATP synthase targeting) (Fig. 2h, i and Supple- mentary Fig. 4a–l). Except for bortezomib, all chemical inhibitors reduced INH$^R$-*katG* to the lower limit of detection for all concentra- tions at or above the DS-parent MIC (Fig. 2h, i and Supplementary Fig. 4a–j). Despite the increased vulnerability of *rpoB*, INH$^R$-*katG* was not more sensitive to growth inhibition or killing by the RNA poly- merase inhibitor rifampicin (Supplementary Fig. 4m, n). In conclusion, isoniazid resistance produces collateral vulnerabilities in diverse bio- logical pathways that are more sensitive to transcriptional and che- mical inhibition under in vitro and host-relevant conditions.

## Alternative redox homoeostasis pathways are more vulnerable to inhibition in INH$^R$-katG

The KatG catalase/peroxidase plays a crucial role in detoxifying hydrogen peroxide ($H_2O_2$) limiting the accumulation of hydroxyl- radicals and subsequent intracellular damage. Consistent with this, INH$^R$-*katG* was (i) more sensitive to inhibition and killing by exogenous $H_2O_2$, menadione, plumbagin, and ascorbic acid (Fig. 3a, b and Sup- plementary Fig. 5a–f); and (ii) had a reduced ability to detoxify reactive intermediates following exposure to $H_2O_2$ (Fig. 3c and Supplementary Fig. 5c, d)[36–38]. We hypothesised that isoniazid resistant *M. tuberculosis* would adapt to the perturbation of KatG activity by (i) upregulating compensatory detoxification mechanisms or (ii) if compensatory pathways were not upregulated, they would be more vulnerable to inhibition. The majority (~88%) of genes within the stress response functional subclass were not differentially regulated; with only the universal stress response proteins TB31.7 (*rv2623*, *rv2624c*, *rv2005c*, *rv2028c*, hspX (*rv2031*), a putative rubredoxin *rv3251c* (*rubA*), and the thioredoxin *trxb1* being upregulated, none of which were more vul- nerable to CRISPRi (Fig. 3d). Interestingly, the *furA-katG-rv1907c* operon was the most highly upregulated (Fig. 3d)[39]. Several well- characterised stress-response genes that were not upregulated were more vulnerable to inhibition, including *sodA* and *trxB2* (Fig. 3d). INH$^R$- *katG* was also more sensitive to auranofin, a chemical inhibitor of the *trxB2* encoded thioredoxin reductase (Supplementary Fig. 6g, h)[40].

Interactions between intracellular iron and $H_2O_2$ lead to the for- mation of hydroxyl radicals[38,41–43]. Given the reduced ability of INH$^R$- *katG* to detoxify $H_2O_2$ we hypothesised that genes involved in iron storage would be more vulnerable to inhibition. Indeed, our WG- CRISPRi screen identified *ideR*, a transcriptional regulator of iron storage[26,44,45], as synthetic lethal in INH$^R$-*katG* (Fig. 3e). Validating this phenotype, unique gRNAs targeting *ideR* required less ATc to impair the growth of INH$^R$-*katG*, reduced the intracellular survival of INH$^R$- *katG* in macrophages, and INH$^R$-*katG* was more sensitive to killing by exogenous FeCl$_3$ compared to the DS-parent (Fig. 3f–h and Supple- mentary Fig. 5i–j). By back-diluting cultures every 5 days into fresh media with ATc and measuring the maximum growth that was reached, the increased vulnerability of *ideR* also reduced the time required to observe a fitness cost in INH$^R$-*katG* (Fig. 3i).

In line with a recent transcriptional study of a *M. tuberculosis katG* mutant, we observed an upregulation of *dosR* and 33 genes (i.e., 65%) in the *dosR* regulon in INH$^R$-*katG* (Supplementary data 3)[46]. Interactions with the redox sensors DosS and DosT allow DosR to regulate genes in response to both redox stress and the transition to hypoxia[47–52]. Based on this, we hypothesised that in addition to increased sensitivity to redox stress, INH$^R$-*katG* would have an altered ability to survive a hypoxic environment. Despite INH$^R$-*katG* consuming oxygen at a comparable rate to the DS-parent and having no reduced viability during adaptation to hypoxia (Fig. 3j), INH$^R$-*katG* had a reduced ability to survive under hypoxia compared to the DS-parent (Fig. 3k). This reduced survivability is consistent with Tn-seq experiments and sug- gests that, like mitochondria, hypoxic conditions generate transient bursts in oxidative stress within *M. tuberculosis*[6,53–56]. In conclusion, WG-CRISPRi screening reveals that pathways which detoxify or limit the production of DNA damaging hydroxyl radicals become more vulnerable to inhibition in *M. tuberculosis* in the absence of the KatG catalase peroxidase.

## Perturbed KatG activity leads to a rewiring of amino acid metabolism in *M. tuberculosis*

Amino acid and nucleotide metabolism were among the functional classes that contained the most genes more vulnerable to inhibition in INH$^R$-*katG* (Figs. 2a, 4a). We hypothesised that these vulnerabilities were the result of metabolic changes needed to adapt to the loss of KatG activity. Using semi-targeted LC/MS to investigate these adapta- tions, we identified 127 metabolite peaks, 17 of which were ≥1.5-fold differentially abundant in INH$^R$-*katG* (Supplementary data 5). We did not observe any increases in the levels of small molecular thiols (CoA, mycothiol, and ergothioneine) that could respond to perturbed KatG activity (Supplementary Fig. 6a, b). However, we did observe a reduction in NADH and an increase in NAD:NADH ratios, although these changes were not statistically significant (Supplementary Fig. 6a). Interestingly, amino acids involved in aspartate metabolism were differentially abundant (i.e., aspartate, lysine, and threonine), which correlated with genes involved in aspartate metabolism being more vulnerable to inhibition (Fig. 4c, d). Methionine was also

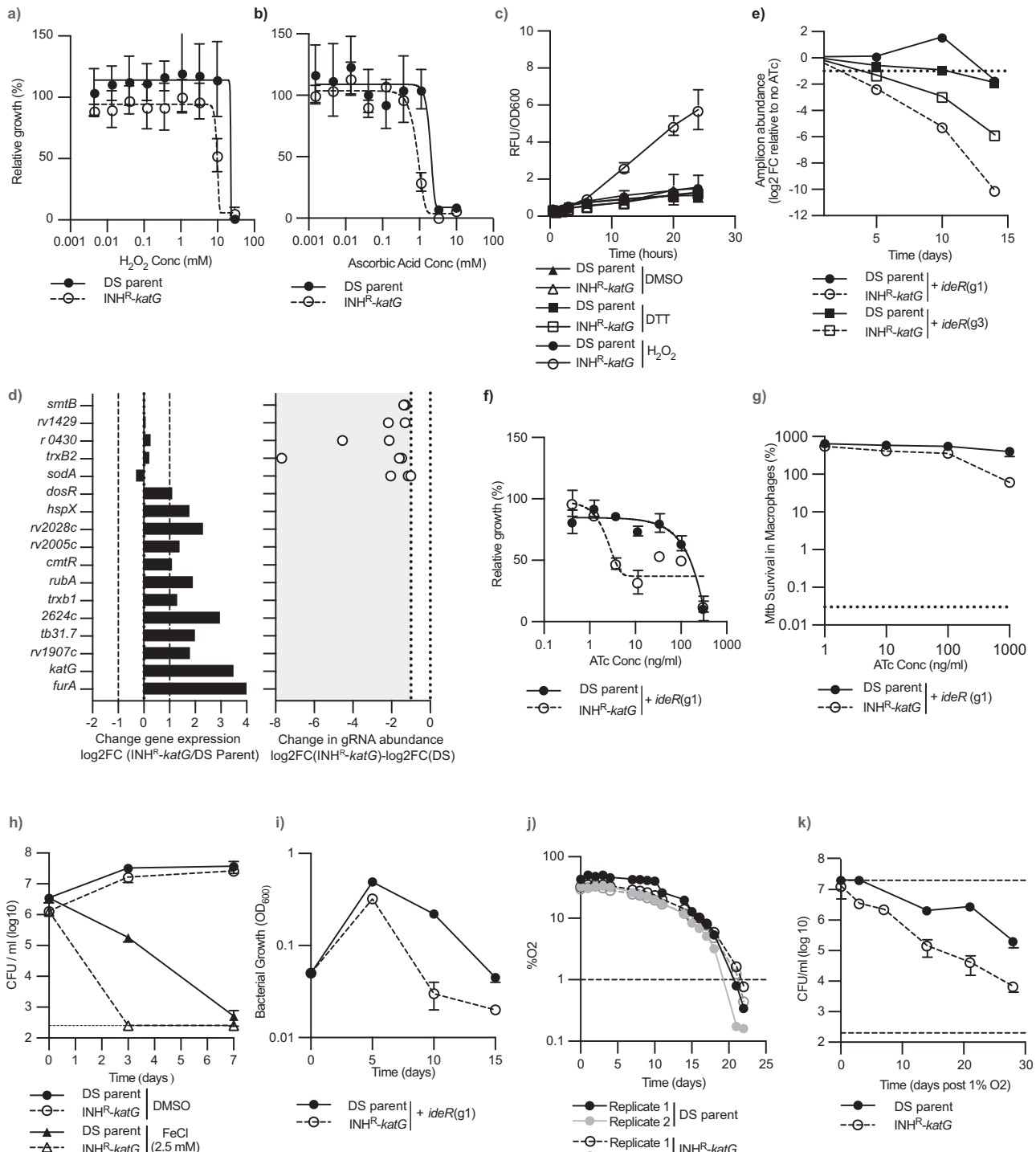

statistically lower in the INH[R]-*katG* relative to DS-parent, but did not meet the 1.5-fold change threshold (Fig. 4b). Although there was no change in the abundance of aromatic amino acids, genes involved in the chorismate (*aroA, B, G, K, F*) and tryptophan synthesis (*trpA, B, C and E*) were also more vulnerable to inhibition in INH[R]-*katG* (Fig. 4d). Aspartate is a precursor to pyrimidine biosynthesis and consistent with this we observed a reduction in N-carbomyl-L-aspartate (i.e., the first committed step in pyrimidine biosynthesis), an accumulation of downstream pyrimidine (CMP) intermediates and genes involved in early steps of de novo pyrimidine biosynthesis were more vulnerable to inhibition (Fig. 4c). Furthermore, the increased accumulation of malate and aspartate, the depletion of oxoglutarate, increase in NAD:NADH ratio and the increased vulnerability of key steps in the

glyoxylate shunt (i.e., *glcB*) suggest that INH[R]-*katG* also reroutes metabolism through the glyoxylate shunt (Fig. 4c). These alterations parallel BDQ treatment and hypoxia-induced metabolic remodelling in *M. tuberculosis* to bypass the oxidative arm of the TCA cycle and reduce electron flow through the electron transport chain there by limiting ROS production[57,58].

Confirming the increased vulnerability of amino acid metabolism, gRNAs targeting methionine (*metA* and *metC*), lysine (*lysA* and *dapE*), or shikimate synthesis (*aroK* and *aroA*) required less ATc to inhibit growth and had an earlier growth inhibitory phenotype against INH[R]-*katG* (Fig. 4e–i and Supplementary Fig. 6). The gRNAs targeting *aroK* and *aroG* also had improved killing against INH[R]-*katG*, whilst all other gRNAs were static against both strains (Fig. 4h and Supplementary

**Fig. 3 | INH$^R$-*katG* utilizes alternative redox detoxification pathways to compensate for the loss of *katG*.** Susceptibility of *M. tuberculosis* DS-parent and INH$^R$-*katG* to growth inhibition by (**a**) H$_2$O$_2$ and (**b**) ascorbic acid (mean ± extrema of two biological replicates, *n* = 3 independent experiments). **c** CellRox based detection of reactive oxygen stress in *M. tuberculosis* DS-parent and INH$^R$-*katG* when exposed to H$_2$O$_2$ (mean ± SD of three biological replicates, *n* = 2 independent experiments). **d** Changes in gene expression INH$^R$-*katG* relative to the DS-parent and increased depletion of gRNA abundance in INH$^R$-*katG* relative to DS-parent. Genes are named using gene name or rv number. gRNAs that are more depleted in INH$^R$-*katG* are presented as white dots, with the positioning of each dot denoting the level of increased depletion on a log2FC scale. Only gRNAs that show a statistically significant depletion on day 14+ ATc-300 are presented. **e**–**h** INH$^R$-*katG* is more sensitive to iron dysregulation**:** (**e**) Abundance of gRNAs targeting *ideR* throughout the WG-CRISPRi screen as detected by amplicon sequencing. Data is presented for cultures exposed to ATc-300. **f** Growth of *M. tuberculosis* DS-parent and INH$^R$-*katG* expressing gRNA targeting *ideR* (mean ± SD of two biological replicates, *n* ≥ 3). The (gx) after each gRNA denotes the specific gRNA targeting *ideR*. **g** Intracellular survival of *M. tuberculosis* DS-parent and INH$^R$-*katG* expressing gRNA targeting *ideR* (mean ± SD of three biological replicates, *n* = 2 independent experiments). **h** Susceptibility of *M. tuberculosis* DS-parent and INH$^R$-*katG* to 2.5 mM FeCl as determined by CFU/ml (mean ± SD of three biological replicates, *n* = 2 independent experiments). **i** Growth kinetics of *M. tuberculosis* DS-parent and INH$^R$-*katG* expressing gRNA targeting *ideR*, Bacterial growth was measured by OD600. All strains were grown in 7H9 media with ATc-300 and back diluted 1/20 with fresh media on days 5 and 10. Data is mean ± SD of three biological replicates, *n* = 2 independent experiments. **j** The ability of *M. tuberculosis* DS-parent and INH$^R$-*katG* to deplete oxygen during the transition to hypoxia was detected using PreSens oxygen sensing spots. Data represents the individual oxygen consumption curves of two biological replicates from a representative experiment. **k** The viability of *M. tuberculosis* DS-parent and INH$^R$-*katG* once the concentration of dissolved oxygen was less than 1% was detected by plating for viable colonies. Data is shown as the mean ± SD of three biological replicates, *n* = 2 independent experiments. Source data are provided as a Source Data file.

Fig. 7a-i). The gRNAs targeting *aroB* and *aroG* had an earlier inhibitory effect on growth that was not observed in ATc dose response assays (Supplementary Fig. 6j-k). Inhibition of *aroK* also impaired the intracellular survival of INH$^R$-*katG* in THP-1 infected macrophages (Fig. 4j).

Consistent with a role in contributing to the detoxification of redox stress, the depletion of *lysA*, *metA* and *aroK* from the DS-parent increased susceptibility to killing by the redox modulators ascorbic acid and plumbagin (Fig. 4k and Supplementary Fig. 7j). We hypothesised that if alterations in amino acid metabolism were an adaptation strategy, then supplementation with exogenous amino acids should protect INH$^R$-*katG* to killing by ascorbic acid. Consistent with this, the addition of exogenous L-lysine and L-threonine increased the viability of INH$^R$-*katG* in the presence of lethal concentrations of ascorbic acid when grown in media without exogenous catalase (i.e., 7H12 + glucose) (Fig. 4l). Whilst the level of protection did not restore viability to the levels of the DS-parent, it increased the viability of INH$^R$-*katG* by >1 log$_{10}$ at multiple time points. This protection was also observed under conditions that contained exogenous catalase in the media (i.e., standard 7H9-OADC) (Supplementary Fig. 7k). The addition of exogenous methionine led to increased rates of killing, whilst aspartate had no effect on viability. In conclusion, perturbed KatG activity in INH$^R$-*katG* leads to metabolic remodelling that produces collateral vulnerabilities in amino acid metabolism.

### Ribosome biogenesis is more vulnerable to inhibition in an INH$^R$-*katG* mutant

The inhibition of protein synthesis increases oxidative stress and cell death due to disruptions in translational fidelity and increases in misfolded protein levels[59]. Consistent with the increased sensitivity of INH$^R$-*katG* to oxidative stress, >30% of genes involved in ribosome biogenesis had increased vulnerability in INH$^R$-*katG* (Fig. 5a). Confirming these increased vulnerabilities, gRNA sequences targeting *rplC* and *rpsP* required less ATc to inhibit the growth of INH$^R$-*katG*, had an earlier growth inhibitory phenotype, and had improved killing against INH$^R$-*katG* under in vitro conditions and within infected THP-1 macrophages (Fig. 5b–e and Supplementary Fig. 8a, b). We hypothesised that if the increased vulnerability of protein synthesis could serve as collateral drug vulnerability, then INH$^R$-*katG* would have increased sensitivity to antibiotics that targeted protein synthesis. Consistent with this, the protein synthesis inhibitors linezolid (oxazolidinone), kanamycin (aminoglycoside) and nitrofurantoin had improved killing against INH$^R$-*katG* (Fig. 5f and Supplementary Fig. 8c–g). Furthermore, linezolid (LZD) had improved killing against INH$^R$-*katG* under in vitro conditions, in time-kill assays and within infected THP-1 macrophages (Fig. 5g, h). When used in combination with INH, subinhibitory concentrations of LZD could also exploit this collateral vulnerability and suppress the emergence of INH resistance (Fig. 5i). In conclusion, INH$^R$-*katG* is more vulnerable to the inhibition of protein synthesis under in vitro and host-relevant conditions, and can be exploited by LZD to suppress INH resistance.

### Collateral vulnerabilities identified in INH$^R$-*katG* translate to clinically relevant genotypes

To determine whether our results would have clinical implications, we first sought to determine whether the collateral vulnerabilities identified in INH$^R$-*katG*, which is an auxotrophic avirulent background, also translated to virulent *M. tuberculosis* strain H37Rv. INH$^R$ strains isolated in H37Rv were tested for increased sensitivity against BDQ and LZD, two drugs that were more efficacious against INH$^R$-*katG* and that have also been FDA-approved for the treatment of drug-resistant *M. tuberculosis*. Of these strains, INH$^R$-KatG$^{M255R}$ was more sensitive to both drugs with a 4-fold increased sensitivity to BDQ and >2-fold shift in sensitivity to LZD. Yet, the INH$^R$-KatG$^{M255R}$ had comparable susceptibility to the DS-strain of H37Rv (Fig. 6a, b).

The INH$^R$-*katG* mutant used in our WG-CRISPRi contains a frameshift mutation at leucine position 458. However, the majority of clinical INH$^R$ via KatG occurs through mutations at a single amino acid, Serine 315, with a swap to either threonine, asparagine or arginine[11]. To investigate if the collateral vulnerabilities identified in INH$^R$-*katG* translated to clinically relevant genotypes, we used mycobacterial recombineering to construct KatGS315 mutants in mc$^2$6206. Whilst we were unsuccessful in constructing KatG$^{S315T}$, we were able to construct a KatG$^{S315N}$ strain (INH$^R$-KatG$^{S315N}$). We also constructed a KatG$^{S315N}$ strain that spontaneously evolved the rifampicin resistance mutation RpoB$^{D435V}$ (MDR-KatG$^{S315N}$), the second most prevalent rifampicin resistant mutation[11]. Both the INH$^R$-KatG$^{S315N}$ and MDR-KatG$^{S315N}$ had increased resistance to INH, although to a lesser extent than the INH$^R$-*katG* mutant (Fig. 6c). Both strains also had increased susceptibility to growth inhibition by BDQ that was comparable to INH$^R$-*katG* (Fig. 6d). Both INH$^R$-KatG$^{S315N}$ and MDR-KatG$^{S315N}$ also had increased susceptibility to killing by BDQ, yet this was not as great as for INH$^R$-*katG* (Fig. 6e). Neither INH$^R$-KatG$^{S315N}$ or MDR-KatG$^{S315N}$ showed increased susceptibility to growth inhibition or killing by LZD (Supplementary Fig. 8h, i). Consistent with our data, INH$^R$ clinical isolates with a KatG$^{S315T}$ mutation have recently been reported as having increased susceptibility to BDQ[46], yet INH$^R$-KatG$^{S315T}$ clinical isolates had no increased susceptibility to LZD (Fig. 6f). In conclusion, collateral vulnerabilities identified in a *katG* frameshift mutant can translate to clinically relevant INH$^R$ genotypes. Additional research is essential to validate these vulnerabilities in alternative clinically relevant INH$^R$ strains.

## Discussion

Therapeutic strategies that exploit the collateral vulnerabilities of drug resistance have the potential to rapidly sterilize and prevent the

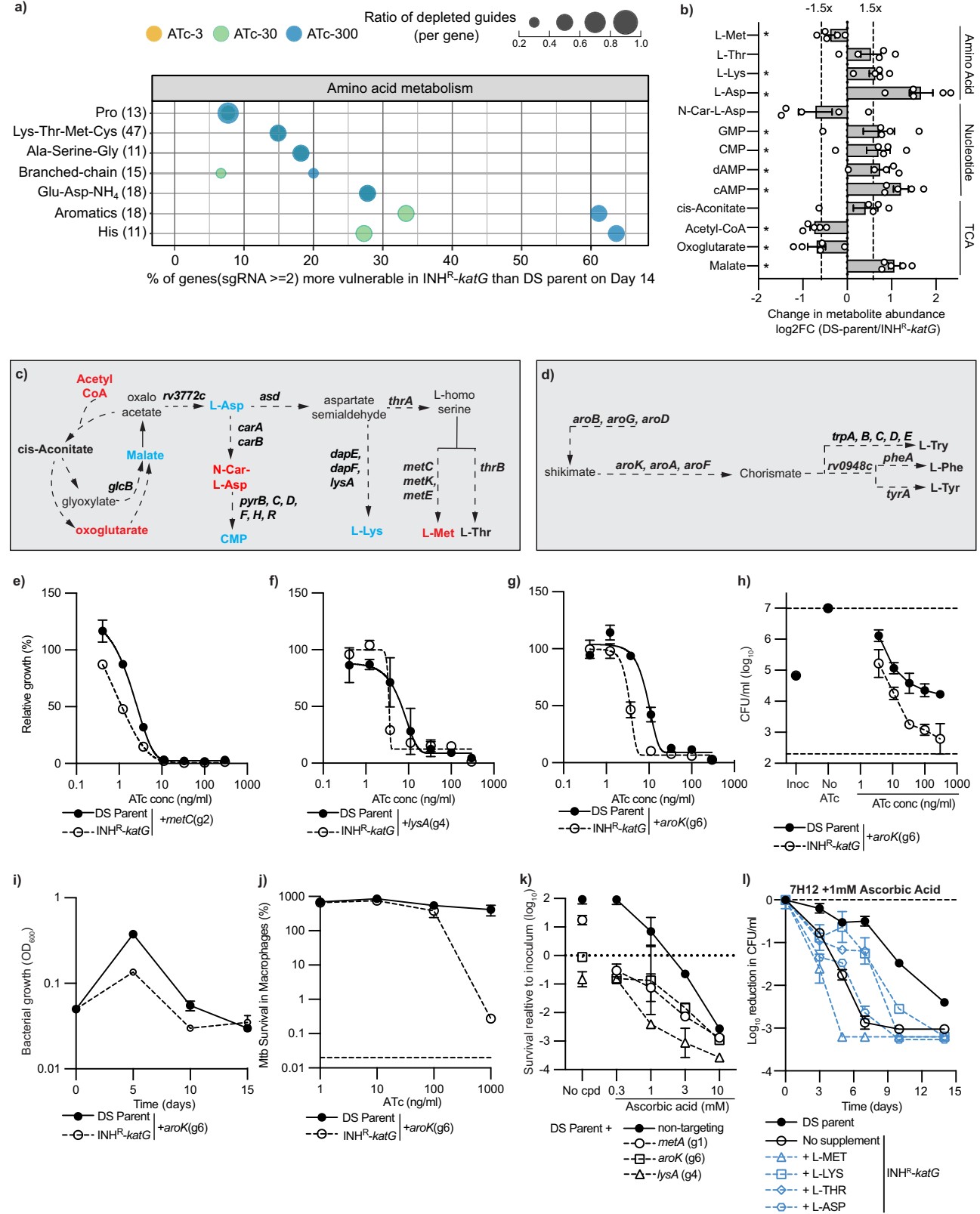

emergence of DR-pathogens. By applying WG-CRISPRi screens with transcriptional and metabolomic approaches we have generated a genome-wide map of collateral vulnerabilities in an INH[R]-*katG* mutant strain of *M. tuberculosis*. This work defines the importance of redox homoeostasis in mycobacterial physiology, uncovers how metabolic remodelling compensates for perturbed KatG activity, and

describes how newly approved TB combination therapies may already be targeting collateral vulnerabilities in DR-strains.

This work identified hundreds of genes from diverse biological pathways that are more vulnerable to transcriptional inhibition in an INH[R]-*katG* mutant. The enrichment of vulnerable genes to pathways that generate oxidative stress when inhibited is consistent with INH[R]-*katG*

**Fig. 4 | Amino acid metabolism is altered in a INH<sup>R</sup>-katG mutant. a** Genes within the amino-acid class with increased vulnerability in INH<sup>R</sup>-katG. Bubble plot represents day 14 data. The x-axis quantifies the proportion of genes with increased vulnerability per subclass. The y-axis shows the names and total number of genes per subclass. Dot size indicates the mean ratio of gRNAs (per gene) more depleted in INH<sup>R</sup>-katG. Dot colour denotes ATc concentrations used. Pro: Proline and 4-hydroxyproline. Lys-Thr-Met-Cys: Lysine, threonine, methionine, and cysteine. Branched-chain: Branched-chain amino acids. Glu-Asp-NH₄: Glutamine, glutamate, aspartate, asparagine, ammonia assimilation. Aromatics: Aromatic amino acid metabolism. His: histidine metabolism. **b** Significantly depleted metabolites are labelled with asterisks ($p < 0.05$ determined by two-sided t-test, $n = 5$ biological replicates). Schematics of (**c**) pathways linking aspartate metabolism with the TCA cycle and nucleotide metabolism and (**d**) aromatic amino acid metabolism. Bolded metabolites highlight those that were detected. Blue, red and black denotes metabolites that are increased, decreased or no-change in INH<sup>R</sup>-katG. Bolded genes highlight those that are more vulnerable. **e**–**g** DS-parent and INH<sup>R</sup>-katG expressing gRNAs targeting (**e**) metC, (**f**) lysA and (**g**) aroK in ATc dose-response assays

(mean ± extrema of two biological replicates, $n = 3$ independent experiments). The (gx) denotes the gRNA name. **h** Viability plots of DS-parent and INH<sup>R</sup>-katG expressing gRNA targeting aroK (mean ± extrema of two biological replicates, $n \geq 3$). Inoc denotes the starting CFU/ml and no-cpd denotes the detected CFU/ml in the absence of ATc. Dashed line represents the minimum detection limit. **i** Growth kinetics of M. tuberculosis DS-parent and INH<sup>R</sup>-katG expressing a gRNA targeting aroK (mean ± SD of three biological replicates, $n = 2$ independent experiments). **j** Intracellular survival of DS-parent and INH<sup>R</sup>-katG cells expressing an aroK gRNA within THP-1 macrophages (mean ± SD of three biological replicates, $n = 2$ independent experiments). **k** DS-parent pre-depleted for metA, aroK and lysA for 5 days was exposed to ascorbic acid. No-cpd is the absence of compound but with 300 ng/ml of ATc. Data is the reduction in viable colonies on day 10, relative to the starting inoculum. Non-targeting gRNA is a negative control. **l** The DS-parent or INH<sup>R</sup>-katG were grown in 7H12 media with 1 mM ascorbic acid and the stated amino acids. Data is expressed as the reduction in viable colonies at each time point relative to the starting inoculum. Source data are provided as a Source Data file.

being more vulnerable to the dysregulation of redox homoeostasis. Importantly, vulnerabilities identified in our WG-CRISPRi screen could be experimentally validated under in vitro and host-relevant conditions. INH<sup>R</sup>-katG also exhibited increased susceptibility to killing by gRNAs and chemical inhibitors that targeted vulnerable pathways. This large increase in killing contrasted with INH<sup>R</sup>-katG having only a small or no increase in susceptibility to growth inhibition (i.e., MIC) to chemical inhibitors. This supports prior observations of oxidative stress being a critical driver of antibiotic lethality and emphasizes the potential of inhibiting redox homoeostasis to potentiate antibiotic lethality[60–65].

This work provides fundamental functional insights into how changes in the physiology of DR-strains generate collateral vulnerabilities. Firstly, disruption of katG activity resulted in an increased reliance on intracellular iron storage and alternative ROS detoxification pathways to maintain redox homoeostasis. Secondly, metabolic rewiring was required to adapt to perturbed katG activity, with exogenous threonine and lysine providing protection against oxidative stress. This suggests that increased aspartate levels reflect an attempt to increase (e.g., lysine) or maintain (e.g., threonine) the production of amino acids that can provide protection to increases in oxidative stress. In Saccharomyces species the decarboxylation of lysine into the polyamine cadaverine allows for uncommitted NADPH to be channelled into glutathione metabolism as an antioxidant strategy[66]. Consistent with this, increases in cadaverine synthesis have been observed in M. tuberculosis INH<sup>R</sup> mutants using gas chromatography-time of flight mass spectrometry (GS-MS)[67]. However, we were unable to identify polyamine peaks in our metabolic spectra, which may have been due to our use of LC-MS rather GC-MS[67]. The enhanced killing of INH<sup>R</sup>-katG when supplemented with methionine is also consistent with prior reports of excess methionine disrupting intracellular thiol pools, potentiating the Fenton reaction and increasing antibiotic killing[68]. Additionally, the increased vulnerability of chorismate metabolism is consistent with recent work showing reduced metabolic flux through this pathway in the absence of KatG[46]. Combined, this work reinforces the fundamental role of amino acid homoeostasis in responding to changes in redox homoeostasis[7,69–75]. A complementary metabolic strategy was also used to favour the use of the glyoxylate shunt to bypass the oxidative arm of the TCA cycle, reduce the flow of electrons through the electron transport chain and thereby limiting ROS by-products[57,58]. Finally, protein synthesis and nucleotide metabolism had the greatest proportion of genes that were more vulnerable to inhibition in INH<sup>R</sup>-katG. Both protein synthesis and nucleotide metabolism have known roles in dysregulating redox homoeostasis and genes involved in these processes have previously been shown to be highly vulnerable in M. tuberculosis[26]. This work extends these findings, highlighting their increased vulnerability in the context of DR-strains

of M. tuberculosis that are experiencing dysregulated cellular physiology and redox homoeostasis.

The majority of clinical INH<sup>R</sup> strains harbour a substitution at KatG Serine 315 to either threonine, asparagine, or arginine. The selection against loss of function mutants is due to a need to preserve at least partial KatG catalase/peroxidase activity to survive within the host environment[76]. By constructing INH<sup>R</sup> and MDR strains that contained a KatG<sup>S315N</sup> mutation, instead of the katG frameshift mutant used in our WG-CRISPRi screen, we were able to confirm that our identified collateral vulnerabilities could translate to clinically relevant genotypes. Importantly, the increased sensitivity of both INH<sup>R</sup>-KatG<sup>S315N</sup> and MDR-KatG<sup>S315N</sup> to BDQ is consistent with recent reports of BDQ hypersusceptibility in INH<sup>R</sup>-KatG<sup>S315T</sup> clinical isolates[46]. Interestingly, none of the INH<sup>R</sup>-KatG<sup>S315N</sup> mutant, MDR-KatG<sup>S315N</sup> mutant, and INH<sup>R</sup>-KatG<sup>S315T</sup> clinical isolates possessed the increased sensitivity to LZD that was observed in our INH<sup>R</sup>-katG mutant. This is likely due to difference in catalase function, with INH<sup>R</sup>-katG likely being a loss of function mutant whilst KatG-S315 mutants retain at least partial catalase/peroxidase activity. These combined findings suggest that the excellent efficacy of the new all oral BPaL (BDQ, pretomanid and LZD) regimen against DR-strains is likely the result of BDQ inadvertently targeting collateral vulnerabilities in DR-strains of M. tuberculosis.

Here we have (i) defined the mechanisms that allow M. tuberculosis to adapt to the perturbation of katG activity and (ii) defined genes and cellular pathways that are more vulnerable to inhibition in isoniazid-resistant cells. In our WG-CRISPRi screen, many of the more vulnerable genes were essential genes and were not detected by transcriptional or metabolic approaches. The translation of this genetic data to existing chemical inhibitors, host relevant models and clinical isolates emphasizes the power of WG-CRISPRi in uncovering novel aspects of mycobacterial biology and highly vulnerable drug targets.

## Methods
### Bacterial strains and growth conditions
The M. tuberculosis strain mc²6206 (H37Rv ΔpanCD, ΔleuCD) is an avirulent derivative of H37Rv[77]. The isoniazid-resistant katG mutant (KatG<sup>L458QfsX27</sup>) used in this study was previously isolated as a spontaneous isoniazid resistant mutant in M. tuberculosis strain mc²6206 where it was named INH-1[20]. Here, INH-1 is renamed INH<sup>R</sup>-katG. M. tuberculosis strain mc²6206 DS-parent and INH<sup>R</sup>-katG were grown at 37 °C in Middlebrook 7H9 liquid medium or on 7H11 solid medium supplemented with OADC (0.005% oleic acid, 0.5% BSA (Sigma A-7906), 0.2% dextrose, 0.085% catalase), pantothenic acid (25 μg/ml), leucine (50 μg/ml) and when required kanamycin (20 μg/ml). Liquid cultures were supplemented with 0.05% tyloxapol (Sigma). Throughout the manuscript, 7H9 refers to fully supplemented 7H9 media, whilst 7H9-K refers to 7H9 supplemented media with kanamycin.

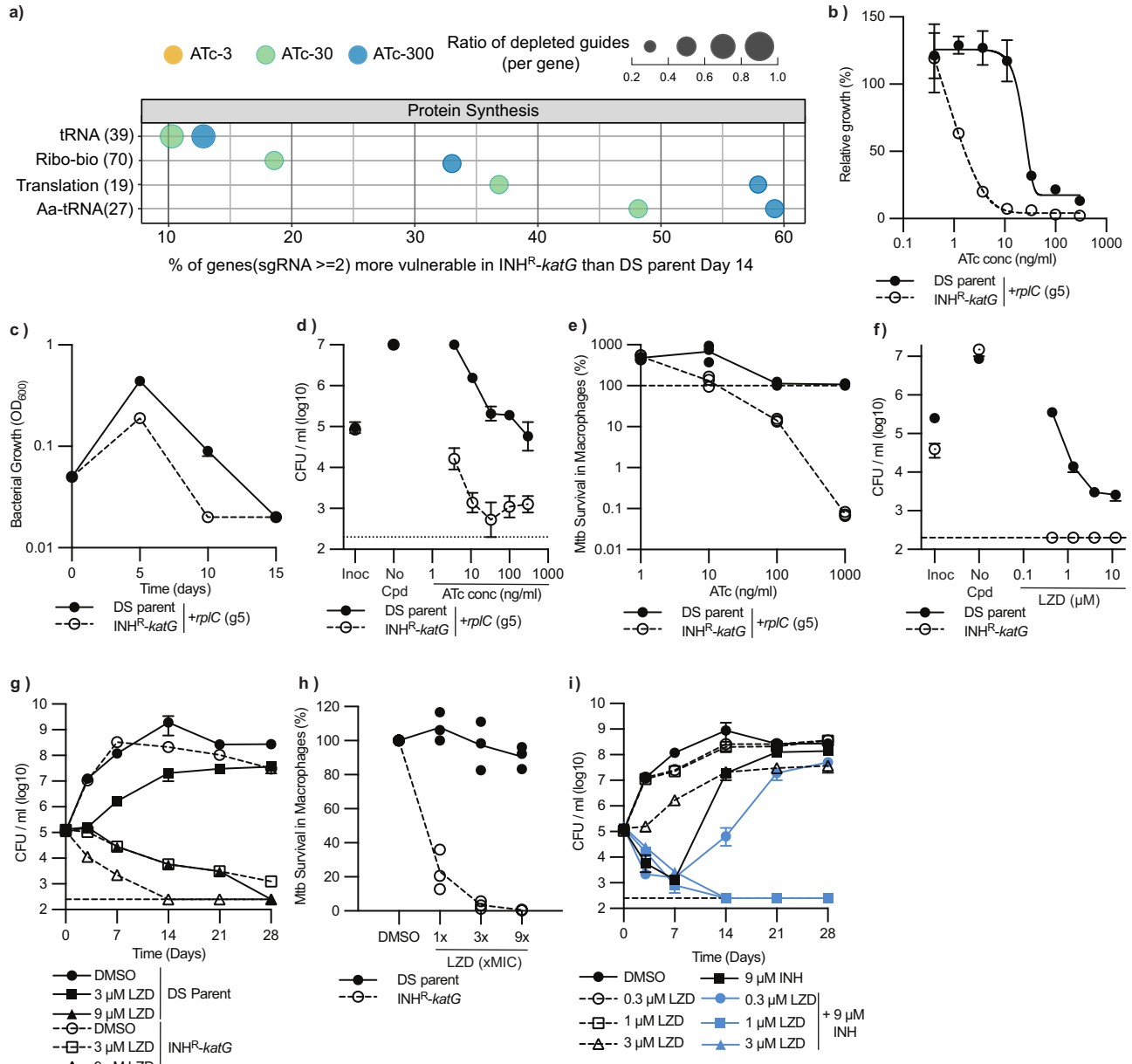

**Fig. 5 | Ribosome biogenesis is more vulnerable to inhibition in an INH^R-*katG* mutant. a** Pathway analysis of more vulnerable genes in INH^R-*katG* within the protein synthesis class. Subclasses are described using classifications from the curated PATRIC database. The bubble plot represents data from day 14. The x-axis quantifies the proportion of genes called "more vulnerable" over the total number of M. tuberculosis genes per functional subclass of the protein synthesis pathway, and the y-axis shows the name of each functional subclass and the total number of genes in each subclass. Within each functional subclass, the dot size indicates the average ratio of gRNAs targeting each gene that is more depleted in INH^R-*katG*. The dot colour denotes the ATc concentration from which the amplicon sequencing was performed. Ribo-bio: Ribosome biogenesis. Aa-tRNA: Aminoacyl-tRNA-synthetases. **b** Growth of *M. tuberculosis* DS-parent and INH^R-*katG* expressing gRNAs targeting *rplC* in ATc dose response assays (mean ± extrema of two biological replicates, *n*=3 independent experiments). The (gx) after each gRNA denotes the specific gRNA targeting each gene. **c**, **d** *M. tuberculosis* DS-parent and INH^R-*katG*

expressing a gRNA targeting *rplC* and assessed (**c**) in continuous log phase growth in 7H9 media with ATc-300, (**d**) for viability in ATc dose response assays and (**e**) for intracellular survival within THP-1 macrophage cells (mean ± SD of three biological replicates, *n* = 2 independent experiments). Data presentation is consistent with Fig. 4 and experiments were performed as described in material and methods. **f**–**h** Susceptibility of *M. tuberculosis* DS-parent and INH^R-*katG* to increasing concentrations of LZD as assessed (**f**) in 96 well plates, (**g**) in time kill assays and (**h**) against intracellular *M. tuberculosis* within THP-1 infected macrophages. For **h**, the MIC of LZD was 1 μM. For (**f**) Inoc denotes the starting CFU/ml and no-cpd denotes the detected CFU/ml in the absence of compound (mean ± SD of three biological replicates, *n* = 2 independent experiments). Dashed line represents the lower limit of detection. **i** *M. tuberculosis* DS-parent grown with INH (9 μM) with or without increasing concentrations of LZD (mean ± SD of three biological replicates, *n* = 2). Source data are provided as a Source Data file.

*Escherichia coli* strain MC1061 was used for the cloning of CRISPRi plasmids. *E. coli* MC1061 was grown at 37 °C in LB or on 1.5% LB-agar supplemented with kanamycin at 50 μg/ml. When used, 7H12 media is 7H9 salts supplemented with cas-amino acids (0.1%), glucose (0.2%) tyloxapol, pantothenic acid (25 μg/ml) and leucine (50 μg/ml)[78].

**Design of *M. tuberculosis* pooled-CRISPRi library**
The pooled-CRISPRi library used in this study was designed by subsetting a published *M. tuberculosis* CRISPRi library that contained 96 K gRNAs, targeting 98.2% of *M. tuberculosis* ORFs[26]. We selected (i) only gRNAs that target the non-template strand of each ORF and (ii) a

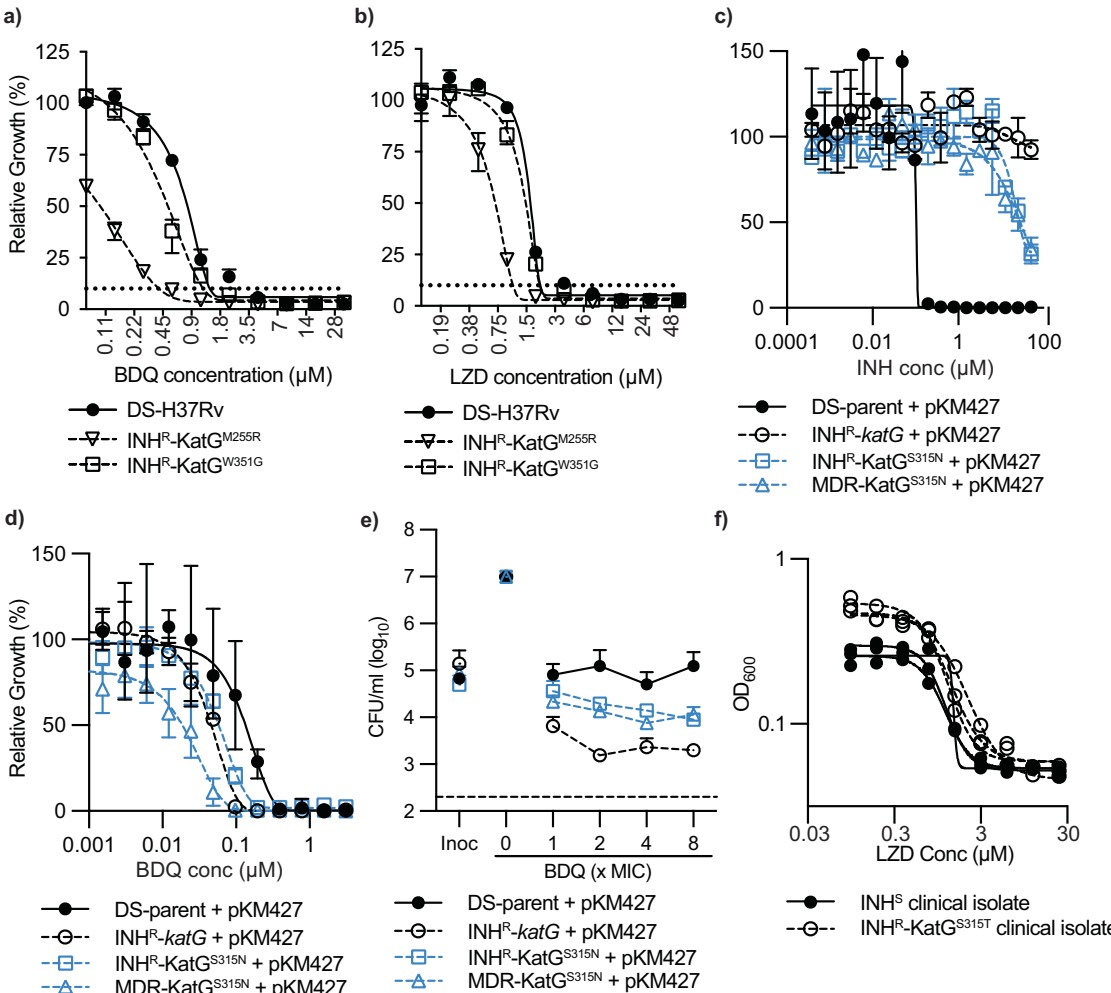

**Fig. 6 | Collateral vulnerabilities translate to clinically relevant INH$^R$ genotypes.**
**a, b** Susceptibility of *M. tuberculosis* H37RV, INH$^R$-KatG$^{M225R}$ and INH$^R$-KatG$^{W351G}$ to inhibition by BDQ and LZD. Growth of M. tuberculosis was determined using REMA assay. **c–e** Susceptibility of *M. tuberculosis* DS-parent, INH$^R$-*katG*, INH$^R$-KatG$^{S315N}$ and a MDR- KatG$^{S315N}$ strain to inhibition and killing by (**c**) isoniazid and (**d, e**) beda-quiline. MIC and MBC assays are mean ± extrema of two biological replicates, *n* = 2 independent experiments. For MBC assays Inoc denotes the starting CFU/ml and

no-cpd denotes the detected CFU/ml in the absence of ATc. The dashed line represents the lower limit of detection. For **c–e** all strains contain the chromoso-mally integrated plasmid pKM427 that is used as part of mycobacterial recombi-neering. The BDQ MIC is 0.2 μM as determined in **d**. **f** Susceptibility of INH sensitive and resistant clinical isolates to LZD The INH$^S$ isolate is TDR-TB 77, whilst INH$^R$ isolate is TDR-TB 42. Both strains are from the TDR-TB strain bank[46]. Source data are provided as a Source Data file.

maximum of six gRNAs with the strongest PAM scores were selected per gene. In total this CRISPRi library contained 22,996 gRNAs that targeted 3991 genes. Ten non-targeting gRNA sequences were inclu-ded as controls. The pooled-CRISPRi library was synthesised and cloned into pLJR965 by Twist Bioscience. Information related to each gRNA in the plasmid pool including gRNA sequence, gene target, abundance in original Twist library, predicted PAM strength, predicted gRNA strength (if available), vulnerability index of target gene based on CRISPRi-Bosch[26], essentiality from prior WG-CRISPRi and Tn-seq screens[26,79], product of each gene if known, strand to which the gRNA binds, position of gRNA along the *M. tuberculosis* mc²6206 genome and gRNA-ID number in the pool is available in Supplementary data 1.

### Transformation and selection of pooled-CRISPRi library into *M. tuberculosis* mc²6206

The constructed pooled-CRISPRi library was transformed into the *M. tuberculosis* DS-parent and INH$^R$-*katG* as follows. From a glycerol stock, 100 μl of each *M. tuberculosis* strain was used to inoculate 10 ml 7H9 media and grown in a T25 flask until confluent. One hundred μl of the outgrowth culture was subcultured into 10 ml 7H9 media and grown until an OD$_{600}$ of 0.4. This was repeated for 12 individual T25 flasks,

giving 120 ml of culture for transformation. At an OD$_{600}$ of 0.4, 1 ml of 2 M glycine was added to each T25 and grown overnight. The 120 ml of culture was harvested in three equal volumes, each washed in 10 ml and again in 5 ml of room temperature 10% glycerol. Pellets were resuspended in a combined volume of 10 ml of 10% glycerol. Two hundred μl of resuspended culture were used to electroporate 5 μl of the constructed pooled-CRISPRi library at a concentration of 50 ng/μl, as previously described[80]. Electroporation's were recovered in 10 ml 7H9 media and grown overnight at 37 °C. A minimum of ten trans-formations were performed to generate a single *M. tuberculosis* CRIS-PRi library. All recovered transformations were harvested and resuspended to an OD$_{600}$ of 0.4 in 7H9-K media. A 200 μl sample from each transformation was used to determine the efficiency of trans-formation via serial dilution and plating. From the remaining culture, 2.5 ml was inoculated into 50 ml of 7H9-K in a T-75 flask, giving a starting OD$_{600}$ of approximately 0.02. Cultures were grown at 37°C until they reached an OD$_{600}$ of approximately 0.5. The proportion of Kan resistant (transformed) cells in the population was determined at the start and end of the outgrowth and compared to a negative control (untransformed *M. tuberculosis*). At an OD$_{600}$ of 0.5, cultures were harvested, individually resuspended in 5 ml volumes, combined, then

the entire library was adjusted to an $OD_{600}$ of 1.0. Cell stocks (1 ml) of the adjusted library were made and frozen at -80°C until required. The total number of Kan-resistant colonies (i.e., transformed) in each library was determined by thawing a single cell stock, performing a tenfold dilution series and plating onto 7H11-K.

## WG-CRISPRi screens

WG-CRISPRi screens were performed in T-25 tissue culture flasks. The screen was initiated by thawing seven 1 ml aliquots of the *M. tuberculosis* pooled-CRISPRi library. The 7 ml of thawed stocks were combined with 7 ml of 7H9-K media to give an $OD_{600}$ of approximately 0.5. To determine the viability of the thawed library, four separate volumes of 100 µl were removed, diluted and plated for CFUs. To outgrow and expand the *M. tuberculosis* pooled-CRISPRi library, 1 ml of the remaining culture was added to 10 ml 7H9-K media in T25 flasks and grown for four days to an $OD_{600}$ of approximately 1. Library expansion was repeated in 12 individual T25 flasks. After four days of expansion, cultures were combined, harvested, and adjusted to an $OD_{600}$ of 1 in 7H9-K media. To determine the effects of CRISPRi-mediated gene repression in the pooled-CRISPRi population, 500 µl of $OD_{600}$ 1 adjusted library was added to 10 ml of 7H9-K media in T25 tissue culture flasks with either 0, 3, 30 or 300 ng/ml anhydrotetracycline (ATc) to induce CRISPRi. Five replicate cultures were started for each ATc condition, with this initial inoculation referred to as the day 0 culture. The remaining $OD_{600}$ 1 culture was spun down in two equal volumes, with the supernatant removed and the cell pellets frozen at −20 °C for future gDNA extraction. All T25 flasks were grown at 37 °C without shaking for 14 days. To maintain log-phase growth, cultures were back diluted using a 1 in 20 dilution on day 5 and again on day 10 into new 7H9-K media with fresh ATc at the same concentration. Cell pellets with the remaining culture were harvested on day 5, day 10, and day 14 via centrifugation at $3200 \times g$ for 10 min. The cell pellets were stored at −20 °C for gDNA extraction.

## DNA extraction, amplicon library construction and sequencing

Genomic DNA was extracted from the frozen cell pellets using Zymo-BIOMICS DNA Miniprep Kits (Zymo Research, #D4300) following the manufacturer's instructions with the following modifications. Briefly, cell pellets resuspended in 750 µl of ZymoBIOMICs lysis solution were lysed by bead beating using a SPEX SamplePrep MiniG 1600 tissue homogenizer (SPEX, Metuchen, New Jersey, USA) at 1500 rpm. The lysis process involved five cycles of 1-min bead beating, with a 20-second pause between each cycle to prevent overbeating. Samples were eluted with 50 µl of nuclease-free water that was preheated to 60 °C. Isolated gDNA was quantified using the Invitrogen Qubit 4 Fluorometer with the Broad Range Qubit kit following the manufacturer's guidelines (Invitrogen, Carlsbad, CA, USA). The integrity of gDNA was determined by running 5 µl of each gDNA sample on a 1 % agarose gel.

To amplify the gRNA sequence, extracted gDNA was diluted to 25 ng/µl and used as a template for PCR amplification. For each sample, 100 ng of gDNA template was used in 50 µl of PCR reaction solution. Master mix was made with Q5 High-Fidelity DNA Polymerase (BioLabs # M0493L) following the manufacturer's instructions. For each strain, 60 individual PCR amplifications were performed (5 replicates of 4 ATc concentrations across 3 days). Each PCR reaction contained (i) one of five forward primers that were offset from each other by a single base to dephase the sequencing library and (ii) a reverse primer with no dephasing. We used a dual indexing strategy with ten forward indexes and 12 reverse indexes. Primer sequences are listed in Supplementary Data 6. PCR conditions were as follows (i) initial denaturation for 2 min at 98 °C, (ii) 20 cycles of 98 °C for 10 s, 62 °C for 30 s, and 70 °C for 20 s and (iii) final extension for 2 min at 72 °C. PCR products confirmed on a 2% agarose gel, were purified using a GFX PCR DNA and Gel Band Purification Kit (Cytiva, # 28903471) and quantified using the

Invitrogen Qubit 4 Fluorometer with the High Sensitivity Qubit kit following the manufacturer's guidelines (Invitrogen, Carlsbad, CA, USA). Each of the 60 purified PCR products per strain was normalized, pooled, and size-selected (245–255 bp) using a Pippin Prep (Sage Science) with 2% agarose dye-free gel cassette (#CEF2010, Sage science, Beverly, USA). The resulting libraries were quantified by Qubit 4 fluorometer (Invitrogen, Carlsbad, CA, USA), then sequenced (100 bp single-end; Illumina NextSeq 2000 at SeqCenter (https://www.seqcenter.com/).

## Deep sequencing data analyses and hit calling

Demultiplexed fastq files were converted to tabular format using SeqKit (v2.3.1)[81]. Sequence features (promoter, gRNA and scaffold sequences) were identified using stepwise fuzzy string matching regardless of the sequencing quality scores (since the expected promoter and gRNA sequences are known, low-quality base calls do not negatively impact downstream analyses). First, only reads with identified promoter regions (upstream of the expected gRNA sequence) were included in subsequent analyses (typically > 99% of raw reads). To account for variable gRNA sequence lengths in the CRISPRi pool, we then used the 34 nt starting from the +1 site as a unique sequence 'tag' to match with the expected gRNA sequence pool (Supplementary Data 1). Only tags with perfect matches to the expected gRNA pool were progressed to generate gRNA count tables for subsequent analyses. Differential abundance tests were performed via the edgeR (v3.42.4) package (classic exact test) to analyse the foldchange of gRNA abundance at each timepoint relative to the corresponding ATc-0 samples[82].

Essential genes were defined when at least 2 gRNAs targeting the same gene of interest had at least a two-fold change in gRNA abundance (Benjamini–Hochberg adjusted $p < 0.01$) relative to the ATc-0 control day 14 with ATc-300. Genes were identified as being more vulnerable to inhibition in $INH^R$-*katG* when the gene of interest was (i) classified as essential in $INH^R$-*katG* and (ii) had ≥2 gRNAs that were depleted in $INH^R$-*katG* by >1-$\log_2$ fold relative to the depletion of the same gRNA in the DS-parent at day 14 with ATc-300.

Functional classification of each target gene was conducted using a manually curated dataset derived from the PATRIC database from the bacterial and viral bioinformatics resource centre[83]. Class and subclass information of each gene was obtained via the subsystems table of the H37rv PATRIC database. In cases where functional classification was missing for a gene, manual classification was applied based on its end products and biological processes.

## Construction and transformation of CRISPRi plasmids that target genes of interest

For validation studies, gRNA sequences of interest from the pooled-CRISPRi library were cloned as individual CRISPRi plasmids, as previously described[31,32,80,84]. Briefly, the gRNA sequence and a complementary sequence were ordered with GGGA and AAAC overhangs. Oligos were annealed and cloned into the CRISPRi plasmid pLJR965 using BsmB1 and confirmed using Sanger sequencing as previously described[80]. All ordered oligos and constructed CRISPRi plasmids are listed in Supplementary Data 7. CRISPRi plasmids were transformed into *M. tuberculosis* DS-parent or $INH^R$-*katG* as previously described[80].

## CRISPRi phenotypic assessment of essentiality and viability in 96 well plates

To determine the consequences of targeted gene repression on bacterial growth and viability, phenotypic assays were performed as previously described[32,80]. Briefly, ATc dose-response assays were performed in 96 well plates. *M. tuberculosis* mc²6206 strains containing CRISPRi plasmids grown in 7H9-K and diluted to an $OD_{600}$ of 0.01 in 7H9-K in a 96 deep well plate. 96 well assay plates were prepared with a 3-fold dilution of ATc along the Y-axis starting at 300 ng/ml of

ATc in row H with a starting inoculum of $OD_{600}$ 0.005. This was achieved by adding 75 μl of 7H9-K to all wells of columns 3–10 except row H, and 113 μl of 7H9-K containing the starting concentration of ATc (i.e., 600 ng/ml ATc) was added to row H of columns 3–10. ATc was diluted along the vertical axis, transferring 37.5 μl between columns, up to row B. Row A was used as a no ATc control. Columns 1, 2, 11 and 12 contained 150 μl of 7H9-K as contamination and background controls. Seventy-five μl of $OD_{600}$ adjusted culture was added to each well to achieve a starting $OD_{600}$ of 0.005. Each column represents the ATc dilution gradient for a single *M. tuberculosis* strain containing a unique CRISPRi plasmid. All experiments included a nontargeting sgRNA (i.e., pLJR965) as a negative control. To assess the fitness costs of gRNAs on growth, duplicate plates were grown at 37 °C without shaking for 10 days. $OD_{600}$ was measured using a Varioskan-LUX microplate reader. $OD_{600}$ reads from duplicate plates relative to the growth of the no-ATc control were analysed using a nonlinear fitting of data to the Gompertz equation(31).

To assess the effects of gRNAs that targeted more vulnerable genes on bacterial viability, duplicate 96 well assay plates were set up as described above. Viability at day 0 was determined using a 4-point ten-fold dilution of the 0.01 diluted culture, with 5 μl of each dilution spotted onto to 7H11-K agar plates. On Day 5, cultures from rows A and D-H were transferred to a new 96 well plate to be diluted. A 4-point ten-fold dilution gradient was performed and 5 μl of each dilution was spotted onto to 7H11-K agar plates. Plates were incubated at 37 °C for 4–5 weeks and colonies were counted.

### CRISPRi phenotypic assessment of essentiality under continuous log phase growth

The increased vulnerability of target genes was also assessed using growth curves, in which the culture was back-diluted to maintain a continuous log phase growth. Experiments were performed by diluting *M. tuberculosis* mc²6206 strains containing CRISPRi plasmids in 7H9-K to an $OD_{600}$ of 0.5. 1 ml of culture was added to 10 ml of 7H9-K in a T25-flasks with ATc-300 to a starting $OD_{600}$ of 0.05. Cultures were grown without shaking at 37 °C. At day 5, the $OD_{600}$ of culture was determined and 0.5 ml of culture was back-diluted into 9.5 ml of 7H9-K in a T25-flasks with ATc-300 and grown without shaking at 37 °C. This was repeated on day 10, with the final $OD_{600}$ being determined on day 15.

### Compound susceptibility and viability assays

The susceptibility of the DS-parent or $INH^R$-*katG* to different compounds was determined using Minimum inhibitory concentration (MIC) assays as previously described[20,85]. Briefly, inner wells (rows B–G, columns 3–11) of a 96-well flat-bottomed microtiter plate (Thermo-Fisher Scientific) were filled with 75 μl 7H9 media. Outer wells were filled with 150 μl 7H9 media as media only controls, and 113 μl of 7H9 media containing compound of interest at the required starting concentration was added to column 2 of row B-G. Compound was diluted 3-fold, by transferring 37.5 μl between wells, down to column 10. Column 11 was kept as solvent only. Strains were diluted to an $OD_{600}$ of 0.01. Seventy-five μl of diluted culture was added to inner wells of the 96-well flat-bottomed microtiter plate containing compound to achieve a starting $OD_{600}$ of 0.005 in a final volume of 150 μl. Plates were incubated at 37 °C for 10 days without shaking. After 10 days, plates were covered with plate seals, shaken for 1 min and the $OD_{600}$ was determined using a Varioskan Flash microplate reader (ThermoFisher Scientific). $OD_{600}$ reads from duplicate plates were corrected for background, and values relative to the growth of the no-ATc control were analysed using a nonlinear fitting of data to the Gompertz equation. Assays to determine bacterial viability in response to compound exposure were set up as described above, with viability determined on days on days 0 and 10. Viability at day 0 was determined using a 4-point ten-fold dilution of the 0.01 diluted culture, with 5 μl of

each dilution spotted onto 7H11 agar plates. At Day 10, culture was removed from appropriate wells and transferred to a new 96 well plate to be diluted. A 4-point ten-fold dilution gradient was performed and 5 μl of each dilution was spotted onto 7H11 agar plates. Plates were incubated at 37 °C for 4–5 weeks and colonies were counted.

### Time kill experiments

Time kill experiments were performed using previously established protocols[20,86]. Briefly, cultures were diluted to an $OD_{600}$ of 0.1 in 7H9 media, with 500 μl added to 9.5 ml 7H9-supplemented media in a T25 flask. 50 μl of diluted compounds were added, with DMSO at a final concentration of 0.5%. In co-treatment experiments, antibiotics were added so the final concentration of DMSO was ≤1%. Culture was removed on stated days, diluted, and spotted as described above for MBC assays to determine the number of viable colonies.

To test if altered amino acid metabolism compensates for the loss of functional *katG*, time-kill experiments were performed in 7H9 with OADC and 7H12 media with glucose as the sole carbon source. To either media type, 100 μl of exogenous amino acids were added to 10 ml media. Amino acids were added from the maximum solubilised concentration (i.e., methionine was added at a final concentration of 20 μg/ml, L-lysine (50 μg/ml), L-threonine (50 μg/ml) and L-aspartate (5 μg/ml).

### Compound susceptibility and viability assays against *M. tuberculosis* strains pre-depleted for genetic targets

*M. tuberculosis* strains depleted for genes of interest using CRISPRi were prepared by diluting log phase culture to an $OD_{600}$ of 0.005 in 10 ml 7H9-K with 300 ng/ml ATc. Cultures were grown without shaking for 5 days at 37 °C to pre-deplete target genes. After 5 days, *M. tuberculosis* expressing a non-targeting control gRNA was diluted 1/10 into 2 ml of 7H9-K+ATc in a deep well 96 well plate to a theoretical $OD_{600}$ of 0.01. As the transcriptional inhibition of *metA*, *lysA* and *aroK* in this study inhibits bacterial growth, 2 ml of undiluted culture was added directly to a deep well 96 well plate at theoretical $OD_{600}$ of 0.01. Assay plates for susceptibility assays were prepared as described above in 7H9-K+ATc. Seventy-five μl of culture from the deep well plate was added to assay plates as described above. Viable colonies were determined on day 0 and 10 as described above. Plates were incubated at 37 °C for 4–5 weeks and colonies were counted. Data is present as the change in CFU/ml relative to the inoculum.

### Hypoxia survival experiments

Oxygen sensing spots (PreSens, Germany) were adhered to the inside of 100 ml glass vials and sterilized before use[87]. Vials were inoculated with 1 ml *M. tuberculosis* adjusted to an $OD_{600}$ of 1 into 29 ml 7H9 for a starting concentration of $OD_{600}$ ~ 0.03. Glass vials were stopped with a rubber stopper to prevent gaseous exchange. Cultures were incubated at 37 °C with shaking (200 rpm). The oxygen concentration was measured following the manufacturer's guidelines by reading the sensor spot through the outside of the flask using a fibre optic cable connected to a Fibox 4 oxygen metre (PreSens). CFU samples were taken using a hypodermic needle that was inserted through the rubber stopper to remove 500 μl culture. Samples were diluted along a four point ten-fold dilution series, spotted onto 7H11-K and incubated at 37 °C. Colonies were counted once visible growth was detected, i.e., approximately 4 weeks.

### CellROX measurements of oxidative stress

CellROX measurements were performed following published protocols[88]. Briefly, mid-log phase cultures of *M. tuberculosis* mc²6206 DS-parent or $INH^R$-*katG* were harvested by centrifugation (3200 × g, 10 min), washed, resuspended in sterile phosphate buffered saline and diluted to an $OD_{600}$ of 1.0. One hundred μl of diluted culture was added to black, clear bottom 96-well microtiter plates (Thermofisher

#165305). Culture was then treated with 1 mM of hydrogen peroxide ($H_2O_2$), dithiothreitol or the solvent control DMSO. Plates were incubated at 37 °C for 1 h. CellROX Green reagent (Thermofisher #C10444) was added to the desired wells at a final concentration of 5 µM and was returned to the incubator in the dark for 30 min. $OD_{600}$ and fluorescence (λEx 485 nm/λEm 520 nm) were measured using a Varioskan Flash microplate reader (ThermoFisher Scientific).

## Macrophage infection assays

THP-1 macrophage infection studies were performed using previously described protocols[35]. Briefly, the human monocytic cell line THP-1 (ATCC Cat# TIB-202) was cultured in standard RPMI 1640 macrophage medium supplemented with 10% inactivated fetal bovine serum and 1 mM sodium pyruvate at 37 °C with 5% $CO_2$. THP-1 monocytes ($5 \times 10^5$ cells/well) were differentiated overnight using 100 ng/ml phorbol myristate acetate (PMA) and seeded in a 24 well-plate. Differentiated macrophages were infected with a mid-logarithmic phase culture of *M. tuberculosis* with or without a CRISPRi plasmid (OD 0.4–0.8) at a multiplicity of infection (MOI) of 10:1 (10 bacteria/1 cell). Infection was allowed to proceed for 1 h. Cells were then washed 3 times with pre-warmed complete RPMI to remove extracellular bacilli. RPMI media containing supplements (pantothenic acid 25 µg/ml and leucine 50 µg/ml), 0.1% BSA and either antibiotic or ATc at varying concentrations were added to the infected cells and incubated at 37 °C with 5% $CO_2$. After 3 days, infected cells were lysed in distilled water containing 0.1% tyloxapol for 5 min at room temperature to determine the number of CFU/ml on 7H11 agar. For strains with CRISPRi plasmids, 7H11 was supplemented with Kan. The percentage of cell viability was determined by normalising CFU/ml counts at day 3 following compound treatment relative to inoculum as determined on day 0.

## Metabolite extraction, mass spectrometry, and semi-targeted analyses

Metabolites were extracted from cultures of both DS-parent and INHᴿ-*katG* as follows. From glycerol stocks, 150 µl of each strain was used to inoculate 10 ml of 7H9 and were grown in T25 flasks at 37 °C without shaking until confluent. One hundred µl of these cultures were then used to inoculate T25 flasks containing 10 ml of fresh 7H9 medium. Once confluent (approximately 2 weeks), these cultures were used to inoculate 6 ml of fresh 7H9 to a density of $OD_{600}$ 0.25 and grown for a further ~48 h ($OD_{600}$ of 0.5-1.2) in T25 flasks at 37 °C without shaking. Culture volumes equivalent to 5 ml of culture at an $OD_{600}$ of 1 were filtered through 0.22 µm filters (Millipore, # GVWP02500) via vacuum filtration. Cell-laden filters were suspended in 2-ml bead beater tubes (SSIbio, # 21276) containing 1 ml of fresh metabolite extraction solvent (2:2:1 ratio of acetonitrile (Sigma-Aldrich, # 900667), methanol (>99.8%), distilled and deionised water (18.2 Ω)) and ~200 µl of 0.1 mm silica beads (dnature, # 11079101z), and cells were lysed by bead beating at 4000 rpm three times for 30 s. Samples were rested for 30 s on dry ice (solid $CO_2$) between each bead beating run. Cell lysates were centrifuged at 13,000 × *g* for 10 min at 4 °C, then ~400 µl of the soluble fraction were transferred to 0.2 µm Spin-X filter columns (Costar, # 8196) and centrifuged at 13,000 × *g* for 3 min at 4 °C. Filtered lysates were transferred to fresh, pre-chilled microcentrifuge tubes before storage at −80 °C. Replicates of the DS-parent and INHᴿ-*katG* were prepared in parallel. Five replicates were prepared in total, with each replicate prepared on separate days. Samples were shipped on dry-ice to Metabolomics Australia (University of Melbourne, Victoria, Australia) for Mass Spectrometry (MS) analysis of the metabolite. Samples were run alongside an in-house standard library containing 550 polar metabolites that were used as references for the assignment of DS-parent and INHᴿ-*katG* sample metabolite peaks, resulting in a semi-targeted approach. Metabolites were separated and detected using a Vanquish Horizon UHPLC system (Thermo Scientific) coupled to an Orbitrap ID-X Tribrid mass spectrometer (Thermo Scientific).

Chromatography conditions were performed as previously reported[89] with modifications. Briefly, separation was performed using a SeQuant *zic*-pHILIC column (150 × 4.6 mm, 5 µm particle size; Merck) at 25 °C, with a binary gradient of solvent A (20 mM ammonium carbonate (pH 9.0; Sigma-Aldrich) and solvent B (100% acetonitrile (Merck, # 100029). The gradient of A/B solvents was run at a flow rate of 300 µl/min as follows: 0.0 min, 80% B; 0.5 min, 80% B; 15.5 min, 50% B; 17.5 min, 30% B; 18.5 min, 5%; 21 min, 5% B; 23–33 min, 80% A. For metabolite detection, the Orbitrap ID-X Tribrid Mass Spectrometer was coupled to a heated electrospray ionisation source and performed as follows: sheath gas flow 40 arbitrary units, auxiliary gas flow 10 arbitrary units, sweep gas flow 1 arbitrary units, ion transfer tube temperature 275 °C, and vaporiser temperature 320 °C. The radio frequency lens value was 35%. Data was acquired in negative polarity with spray voltages of 3500 V. Samples were run in a random order, and the quality of data produced was assessed by the peak variation of pooled samples (all samples combined equally) and four internal standards ($^{13}C_5$,$^{15}N_1$ Valine, $^{13}C_6$ Sorbitol, $^{13}C$,$^{15}N$-UMP, $^{13}C$,$^{15}N$-AMP) that were added to each sample. The data was collected using Thermo Tracefinder (V 4.1) (General Quan Browser). Metabolites were assigned to sample peaks in El-Maven v.0.12.1 by comparison to the peaks in the standard library. WT and INHᴿ-*katG* peaks that were also observed in metabolite extraction solvent-only samples (at >20% of the WT) were excluded. Identified metabolites of the DS and INHᴿ-*katG* samples were provided as raw values from the area under the curve. Raw data were processed and analysed using the MetaboAnalyst v5.0 web server (https://www.metaboanalyst.ca/docs/About.xhtml)[90]. Peak intensities of all identified metabolites in a combined dataset of DS-parent and INHᴿ-*katG* were median normalised and compared between DS and the INHᴿ-*katG* groups using the Statistical Analysis [one factor] module.

## RNA extraction and analysis of RNA sequencing

Five replicate cultures (of both DS-parent and INHᴿ-*katG*) were inoculated into 7H9 media at a starting $OD_{600}$ of 0.1 in T25 tissue culture flasks, grown for 3 days at 37 °C without shaking, then harvested and RNA extracted as previously described[80]. Briefly, the volume of culture harvested was determined as follows ($OD_{600}$ x volume of culture(ml) = 2.5). Harvested pellets were resuspended in 1 ml TRIzol, bead beaten in a 2 ml tube with 200 µl of 0.1 mm Zirconia/silicon beads for 3 cycles of 30 s at 4800 rpm followed by 30 s on ice and frozen overnight at −20 °C. Frozen samples were thawed, mixed with 0.2 ml chloroform and centrifuged in Invitrogen PhaseMaker tubes (Cat No: A33248) for 15 minutes at 12,000 g. The clear, upper aqueous phase was transferred to a clean 1.5 ml Eppendorf tube and mixed with an equal volume of ethanol. RNA was extracted using Zymo-RNA Clean and Concentrator (Cat No: R1019) and DNase treated using Invitrogen Turbo DNA-free kit (Cat No: AM1907). Removal of DNA was confirmed by PCR using 1 µl of extracted RNA as a template with the primer combination of MMO200 + MMO201. Extracted RNA was quantified and quality controlled using the Aligent 2100 Bioanalysis system following the manufacturer guidelines. RNA was prepared using GenTegra RNA tubes (#GTR5025-S) and shipped at room temperature to SeqCenter for RNA sequencing.

Each replicate was sequenced using the 12 M Paired End rRNA depletion RNA sequencing service. Library preparation was performed with Illumina's Stranded Total RNA Prep Ligation with Ribo-Zero Plus kit and 10 bp IDT for Illumina indices. Sequencing was done on a NextSeq2000 giving 2 × 51 bp reads. Demultiplexing, quality control, and adaptor trimming were performed with bcl-convert (v3.9.3). Adaptor removal and quality trimming were conducted with bbduk (a part of the BBTools suite bbmap v39.06 https://sourceforge.net/projects/bbmap/). In all cases, low-quality reads with a Phred quality score <10 were removed. Contaminant removal was then carried out to filter out all reads that have a 31-mer match to PhiX (a common Illumina

spikein) allowing one mismatch. After pre-processing, 13,719,397 read pairs on average per sample were used for downstream analyses.

The cleaned paired-end transcriptomic FASTQ files were then aligned to the *Mycobacterium tuberculosis* mc²6206 complete genome (NCBI Accession number: PRJNA914416) with Bowtie2 using the default settings (v2.4.5)[91]. The output alignments were saved as SAM files, converted to sorted BAM files, and produced index BAI files with SAMtools (v1.16.1)[92]. The resulting alignment files (i.e., BAM and BAI files) were loaded in R (v4.3.0) with the package Rsamtools (v2.16.0)[93,94]. Gene counts were calculated using packages GenomicFeatures (v1.52.0) and GenomicAlignments (v1.36.0). Differential expression of each gene was calculated with DESeq2 (v1.40.1) between DS-parent and INH$^R$-*katG* strains. Gene expression with at least 2-fold difference (Benjamini–Hochberg adjusted $p < 0.05$) between DS and INH$^R$-*katG* were considered to be differentially expressed.

### qPCR of CRISPRi knockdown strains

*M. tuberculosis* strains expressing sgRNA were inoculated at a starting OD$_{600}$ of 0.1 in 10 ml 7H9-K media with variable levels of ATc and grown for three days and harvested. RNA was extracted, cDNA synthesis and qPCR experiments were performed as previously described[80]. Primers used in qPCR experiments are listed in Supplementary Data S6. Ct Signals were normalized to the housekeeping *sigA* transcript and quantified by the comparative Ct method ($2^{-\Delta\Delta Ct}$) method. Error bars are standard deviation of three technical replicates.

### Mycobacterial recombineering

*M. tuberculosis* KatG point mutants were constructed using mycobacterial recombineering as previously described[95]. Briefly, *M. tuberculosis* strain mc²6206 DS-parent was transformed with pKM427 (Zeo$^R$ with a broken Hyg$^R$ cassette for selection) and pKM461 (Kan$^R$). *M. tuberculosis* with pKM427 and pKM461 were grown and made electrocompetent as described above, but with the addition of ATc to induce the Che9C annealase. Electrocompetent cells were transformed with (i) an oligo to introduce the *katG* SNP of interest and (ii) the MMO591 oligo to repair the broken Hyg$^R$ cassette on pKM427. Oligos are listed in Supplementary Data 6. Transformed cells were plated onto 7H11 containing OADC-Leu-Pan and hygromycin (50 μg/ml) to select for cells that had active recombination. The resulting colonies were grown in 24 well plates containing 1 ml of 7H9 media and hygromycin. Colonies that grew were screened for isoniazid-resistance in 96 well plates as described above. Genomic DNA was extracted from strains that were isoniazid resistant and mutations were confirmed by whole genome sequencing[20]. Colonies with the confirmed mutation were cured of pKM461 by growing for approximately 2 weeks in 7H9 media, diluted and plated onto 7H11 containing OADC-Pan-Leu and 3% sucrose. Resulting colonies were grown in 24 well plates, with colonies that grew being screened for a loss of kanamycin resistance in 96 well plates as described above.

### Isolation and characterisation of isoniazid resistant *M. tuberculosis* H37Rv

Isoniazid (INH) resistant *M. tuberculosis* (H37Rv) strains were obtained by plating onto agar that contained 16 to 32x MIC$_{90}$ of INH. Individual colonies resistant to INH were grown, and a PCR specific for the *katG* coding region was conducted. Sanger sequencing was utilised to determine causative SNPs on these PCR products. A Resazurin reduction microplate assay (REMA) assay was conducted on strains with the M255R and W351G mutations to determine the MIC$_{90}$ of these strains to INH, BDQ and LZD relative to the drug-susceptible parent H37Rv strain[96].

### Statistics & reproducibility

No statistical method was used to predetermine the sample size. One DS-parent replicate out of five taken on day 5 ATc-0 was destroyed due to contamination, and thus was not included in the study. One of the six replicates in the metabolomic data was identified as an outlier and was excluded from the analysis. For experimental validations, a small number of replications (two to three) were used mainly due to the practicality of allowing horizontal comparisons within the same experimental setups. The experiments were not randomized. The Investigators were not blinded to allocation during experiments and outcome assessment.

### Reporting summary

Further information on research design is available in the Nature Portfolio Reporting Summary linked to this article.

## Data availability

The amplicon sequencing and RNA sequencing data that support the findings of this study have been deposited in NCBI Sequence Read Archive with the BioProject accession code PRJNA1041353. The Metabolomic data generated in this study have been deposited in the MetaboLights database under accession code MTBLS11277. Manually curated pathway calls were derived from the PATRIC databases [https://www.bv-brc.org/search/?and(keyword(Mycobacterium), keyword(tuberculosis),keyword(H37Rv))]. Source data are provided with this paper.

## Code availability

The code used in this study is available from GitHub [https://github.com/Cecilia-Wang/2023_CRISPRI][97].

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

## Acknowledgements

This research was supported by the Health Research Council of New Zealand via project grant 20/459. We thank Sir Charles Hercus Health Research Fellowship grant 22/156 (M.B.M.) and 23/228 (S.A.J.). We acknowledge the Agilent Early Career Professor Award (J.H.Y.) and the National Institutes of Health via grants NIH U19 AI62598 (J.H.Y.) and NIH R01 AI146194 (J.H.Y.). A University of Otago Doctoral Scholarship supported N.J.W.; We thank members of the Department of Microbiology and Immunology 6th floor for helpful discussions. We thank Peter Finin (NIAID) for providing recombineering plasmids and assistance with mycobacterial recombineering protocols.

## Author contributions

M.B.M. and S.A.J. conceived the project. X.W., N.J.W., W.J., C.J.S., H.R.K., C.Y.C., M.C., N.E.S., B.O.A., E.S., A.P., S.A.J. and M.B.M. performed the experiments. X.W., W.J., B.N., N.P.W., J.H.Y., G.M.C., P.C.F., S.A.J. and M.B.M. interpreted the data. X.W. and M.B.M. wrote the manuscript with input from all other authors.

## Competing interests

The authors declare no competing interests.
