## [Peer Review File · Nature Communications]

REVIEWER COMMENTS

Reviewer #1 (Remarks to the Author):

The study by Wang and colleagues shed lights on the key metabolic pathways that can be targeted in INH resistant *Mycobacterium tuberculosis*.

By using an innovative CRISPRi screen combined to conventional microbiology, RNA analysis and metabolomics, the authors have identified key metabolites and pathways that could be targeted to revert INH resistance.

The study is extremely timely as nay research in this area could lead to major breakthrough in tackling AMR and Tuberculosis. #

The manuscript is very clear and well written. The experiments have been conducted diligently with appropriate controls and statistical analysis.

However, few concerns are raised as below:

1) As KatG is involved in REDOX homeostasis, the authors should consider measuring mycothiol/mycothione; NAD/NADH and AXP pool (ATP, ADP, AMP). This information will clearly define the impact of the proposed pathways into Mtb susceptibility to INH and redox.

2) Although in Figure 4 the authors proposed that altered amino acids metabolism compensate the loss of functional katG, there is no direct correlation between those 2 observations. The authors should consider testing that hypothesis by supplementing culture medium with for example aspartate and measure MIC in the strains used in this study. That will be indicative of a direct correlation between the level of intracellular level of amino acids and katG inhibition for example.

Reviewer #2 (Remarks to the Author):

Wang et al. use CRISPRi knockdowns to generate a map of vulnerabilities in an INH-resistant katG mutant of Mtb. They identify numerous vulnerabilities of a katG frameshift mutant. This is a very interesting approach that could serve as an example of how specific vulnerabilities of drug-resistant Mtb could be targeted for better therapy. The paper is broad rather than deep, but provides a mountain of potentially interesting data to mine.

Things for the authors to consider:

1. As the authors note in the Discussion, the clinical implication of their findings may be limited by their

use of a katG frameshift mutant, when by far the most common INH-R katG mutant is S315T. The impact of the findings of this paper would be greatly strengthened if they could validate at least some of their findings in katG-S315T strains. Figure 5J is not convincing in this respect. The authors should test some of their hit genes in katG-S315T strains relative to the DS parent strain and see if the phenotypes observed in the katG frameshift mutant are also true in S315T.

2. Related to point 1 above, the paper would benefit from some small, confirmatory studies in proper, pathogenic Mtb. For example, some of the hits the authors follow up on focus on metabolic genes. Given that they work with a panleu auxotroph, it is unclear how this genetic background might impact some of the hits they follow up on.

3. The implicit assumption from the authors is that the level of knockdown at different ATc concentrations is the same between the DS and the INH-R strain. Given that this is a new genetic knockdown approach this should be validated.

4. In Figure 1G, 388 hits genes seems like a lot (almost 10% of all Mtb genes). How many of these hit genes are essential in the DS strain? I would be curious how many essential genes come up as hits in their screen.

5. The authors define genes as more vulnerable if they had 2 or more gRNAs that depleted more than 2 fold. How many genes had 2 or more gRNAs that enriched by more than 2 fold? How many genes had guides that both depleted and enriched?

6. There seems to be no correlation between how the gene behaved in the screen and the fold reduction in the ATc MIC (Figure 2B). How do the authors interpret this?

7. The MIC experiments in Figure 2 and Ext Data Figure 3 are somewhat difficult to interpret with respect to the screen. In the screen, there are no hits at ATc-3 ng/ml. But in their follow-up, the MIC differences that the authors see between DS and INH-R tend to be at ATc concentrations below 10 ng/ml. For example, Figure 2C would suggest that gyrB hit at the ATc-3 screen, but perhaps even not a hit at higher ATc concentrations (ATc-30 and ATc-300). How do the authors explain this?

8. Related to point 6 above. A negative control gene ATc MIC experiment would be nice to include the main text. Some of the differences between DS and INH-R are very subtle, so it would be good know what a non-hit control gene looks like in this assay.

9. The difference in the extent of kill between DS and INH-R is impressive (Fig 2H,I; Ext Fig 4). In many cases it seems as if the DS plateaus at a higher viable CFU than INH-R at higher drug concentrations. It would be good to also show data for MICs for these drugs. Is the difference the authors see due to decreased tolerance in INH-R? Decreased persistence? Something else?

10. The authors propose that INH-R remodels metabolism to compensate for perturbed KatG activity. This conclusion is quite interesting but is not clear that the authors demonstrate it to be true. That they see differences in metabolite levels is not necessarily demonstration that this is "compensation." Compensation implies that there is a fitness benefit to such a change, which is not consistently borne out in their data.

11. The connection with ribosome biogenesis is also quite interesting. Does knockdown of rplC or rpsB lead to ROS in the DS and INH-R strain? Do linezolid and kanamycin increase ROS levels in their strains?

12. Figure 1G, it is not clear how these genes were called as hits. More explanation in the Figure Legend would be welcomed. This figure also doesn't convey what are the strongest hits from the screen. Is there an alternative way to display the screen to show both effect size and statistical significance, e.g. a volcano plot?

13. Figure 2A (and elsewhere) is labelled as "pathway enrichment analysis." It is not clear from the figure

how this analysis quantifies enrichment. Please explain. Is the enrichment statistically significant? What is the null hypothesis?

Reviewer #3 (Remarks to the Author):

Summary: Wang et al. report the generation of a whole genome CRISPRi library targeting *Mycobacterium tuberculosis* (Mtb) and its use to probe for collateral vulnerabilities that emerge in INH-resistant strains associated with mutations in the KatG catalase-peroxidase. Initial characterization of the CRISPRi library served to validate predictions of gene essentiality from a prior Mtb genome-wide CRISPRi screen reported by Bosch et al. By employing a multi-disciplinary approach consisting of CRISPRi screening in an INH-resistant *katG* mutant background couple with transcriptomics and metabolomics, they also identified multiply genes and pathways that exhibit enhanced vulnerability to inhibition (by transcriptional silencing) in a *katG* mutant vs. wild-type. These included genes involved in redox homeostasis, amino acid metabolism, and ribosome biogenesis. It is hypothesized that metabolic and transcriptional adaptation to compensate for the loss of KatG activity lead to “druggable vulnerabilities” that may offer novel strategies for treatment of drug-resistant Mtb. The potential for exploiting these vulnerabilities was highlighted by demonstration of improved potency of multiple classes of protein synthesis inhibitors against INHR *katG* mutants, both in vitro and in a macrophage infection model. This observation was further extended by analysis of pre-existing data on ~12,000 Mtb clinical isolates from the CRyPTIC database that revealed that a greater proportion of INHR strains had lower MICs for linezolid than INHS strains. Overall, this was a well-written and clearly organized manuscript with figures that effectively conveyed key aspects of a large dataset. Methods were, for the most part, described in sufficient detail and supplementary data was also very relevant. One significant weakness or concern relating to experimental design plus a few minor corrections/suggestions are detailed below.

Major:

- The use of a *katG* frameshift mutant, rather than more clinically relevant S315T mutants (>70% of INH-R clinical isolates) introduces undesirable caveats into the interpretation of the data. While this is briefly acknowledged in the discussion (line 355), the implications of this choice should be elaborated on in more detail. S315T KatG has only mildly diminished catalase-peroxidase activity compared to *katG* null genotype expected in frameshift mutant. Given the modest level of impact of CRISPRi silencing of “hit” genes in the *katG* null mutant, the enhanced vulnerability of at least select target genes should be validated in the S315T *katG* background.
- Statistical analysis of quantitative phenotypic data (e.g. MICs) were not included. For example, the authors claimed that when gene *isc* was targeted there was impairment in intracellular survival (line 167-68, Extended Data Fig. 3n). However, the figure shows that both DS-Parent and INHR-*katG* had survival above 100% and only a marginal difference is evident. Was any statistical analysis done to validate the difference?

Minor:

- Line 40 – Suggest updating reference to six-month regimen to recently approved 4 month RPT-MOX

regimen for drug-susceptible TB.

- Line 50-51 – Either here or in discussion, it would be helpful to elaborate on the types and frequency of clinically relevant mutations in *katG* and their different impacts on catalase-peroxidase activity, fitness, and virulence phenotypes (see major concern above).
- Line 103-104 – Do not think that “disagreed essential genes” is grammatically correct. Perhaps “discrepant” or “genes with discrepant essentiality predictions”
- Line 125 – Should this read “depleted by >1 log₂ fold change MORE than observed in the DS-parent”?
- Line 218 – The statement that “DosR regulates survival under oxidative stress” does not seem to accurately reflect the role of DosR under hypoxic stress (basically the opposite). This would typically mean that for e.g. a *dosR*- mutant would be hypersusceptible to H₂O₂ or other oxidants, and to my knowledge that is not the case. Please clarify this statement.
- Line 372-373 – The double use of translation in “The translation of this genetic data translated to existing chemical inhibitors” is unnecessary.
- Line 375-376 - What is a “portable CRISPRi platform”? Given the species-specificity of promoters, it seems unlikely that a single platform would work across diverse pathogens.
- Line 394: Please clarify what is meant by “functional 1.5% agar”.
- Maintain consistency with either number or word usage at the beginning of a sentence (100µl in Line 640 and One hundred µl in Line 671).
- Line 765-66: Please provide a link to scripts in GitHub.
- The number of replicates of each experiment, particularly the phenotypic characterization of single-gene CRISPRi knockdowns, could not be determined. Please clarify the number of technical and biological replicates for each assay type.
- Line 1097 – Check grammar “genes that are of more vulnerable” should read “genes that are more vulnerable”.
- Figure 3 legend (lines 1148-1153) – There is a panel K but no description of (k) in the legend. Also, the legend for (i) and (j) seems to correspond to panels J and K, respectively. Also, (f) and (g) leading into description of these panels are not bolded, whereas letters for other panel descriptions are.
- Line 1185 – Should be “at the stated concentration”. Also, should (m) and (n) be bolded?
- Providing additional details such as MIC values about certain experiments would be helpful. Specifically, in Figure 5 Linezolid was used as 1X-9X MIC in macrophage survival assay without mention of MIC value in graphs or in text section. This info could either be integrated into the graphs or as a supplementary table or in some other way.
- Convert MIC values in either µg/ml or µM for consistency and quick comparison of LZD sensitivity between INHR-*katG* and clinical isolates. LZD in µg/ml was used in assays with clinical isolates whereas in other assays it was in µM concentration (Fig.5).
- Maintain consistency in italicizing scientific names in reference section.
- Should in-text citations be as underlined superscript numbers as per journal recommendation e.g. 34 instead of (34)? Maybe this is done in proof stage.
- Could not find the RNA sequencing data using provided accession number (PRJNA1041353). If this bioproject is locked until time of publication, can disregard.
- Spell checks needed.
- Line 468: (Quibt)
- Line 403: (pJLR965)
- Line 610 (CRISRPi)

- Line 750 (fremoved)
- Remove an extra period (Line 1146) and add a period (Line 634).
- Maintain consistency with either number or word usage at the beginning of a sentence (100µl in Line 640 and One hundred µl in 671).

Reviewer #4 (Remarks to the Author):

Reviewer comments in black
Our responses are in blue.

REVIEWER COMMENTS

Reviewer #1 (Remarks to the Author):

The study by Wang and colleagues shed lights on the key metabolic pathways that can be targeted in INH resistant *Mycobacterium tuberculosis*.

By using an innovative CRISPRi screen combined to conventional microbiology, RNA analysis and metabolomics, the authors have identified key metabolites and pathways that could be targeted to revert INH resistance.

The study is extremely timely as nay research in this area could lead to major breakthrough in tackling AMR and Tuberculosis. #

The manuscript is very clear and well written. The experiments have been conducted diligently with appropriate controls and statistical analysis.

However, few concerns are raised as below:

R1Q1: As KatG is involved in REDOX homeostasis, the authors should consider measuring mycothiol/mycothione; NAD/NADH and AXP pool (ATP, ADP, AMP). This information will clearly define the impact of the proposed pathways into Mtb susceptibility to INH and redox.

We agree with the reviewer, that as KatG is a major detoxifier of ROS, it was conceivable that perturbed KatG function in INH^R-*katG* could alter intracellular levels of NAD/NADH and/or low molecular weight thiols (e.g., CoA, mycothiol, and ergothioneine) that also contribute to ROS detoxification.

We observed no statistically significant differences for low molecular weight thiols in the semi-targeted MS data or when analysing raw MS spectra (Fig below). Peaks for ergothioneine and mycothiol were predicted based on published literature. However, we did observe a reduction in NADH and an increase in NAD/NADH ratio in INH^R-*katG*. Whilst this increase was not statistically significant, it is in line with our observation that *M. tuberculosis* responds to perturbed KatG function through metabolic rewiring.

We have included the figure below as Extended data Figure 6 and the following sentence to the results of the revised manuscript lines 271-274

“We did not observe any increases in the levels of small molecular thiols (CoA, mycothiol, and ergothioneine) that could respond to perturbed KatG activity (Extended data Fig. 6a-b). We did observe a reduction in NADH and an increase in NAD/NADH ratios, although these changes were not statistically significant (Extended Data Fig. 6a).”

- a) Extended Data Fig. 6a
- b) Extended Data Fig. 6b

R1Q2: Although in Figure 4 the authors proposed that altered amino acids metabolism compensate the loss of functional *katG*, there is no direct correlation between those 2 observations. The authors should consider testing that hypothesis by supplementing culture medium with for example aspartate and measure MIC in the strains used in this study. That will be indicative of a direct correlation between the level of intracellular level of amino acids and *katG* inhibition for example.

To test this correlation we have performed experiments to determine if the addition of altered amino acids (i.e. methionine, lysine, threonine or aspartate) could protect INH^R-*katG* from killing by ascorbic acid. We focused on ascorbic acid, as the increased sensitivity of INH^R-*katG* to ascorbic acid was reported in our original submission (Fig 3b and extended data 5e).

We performed time-kill experiments in both 7H9-OADC media and 7H12 with glucose as the sole carbon source. The primary difference between these media types is that 7H12 is composed of 7H9 salts with casamino acids, but does not contain the additional catalase that is present in OADC (C=catalase). We hypothesized that the absence of catalase would amplify the lethal effects of ascorbic acid. Consistent with this, ascorbic acid had increased lethality against both the DS_{parent} and INH^R-*katG* when grown in 7H12 media compared to 7H9 (Figure 4l and Extended Data Fig. 7k in the revised manuscript). In these experiments the addition of both lysine or threonine was able to provide protection to INH^R-*katG* in both media conditions (Figure 4l and Extended Data Fig. 7k in the revised manuscript). Whilst the level of protection did not restore it to the levels of the DS_{parent}, it increased viability of INH^R-*katG* by >1 log₁₀ at multiple time points. The protection provided by exogenous lysine is consistent with depletion of *lysA* in the DS_{parent} having the greatest effect on viability against ascorbic acid compared to either *metA* or *aroK* depletion (Fig 4k of original submission). Interestingly, under both conditions the addition of exogenous methionine increased rates of killing, reaching the lower limit of detection earlier than the non-supplemented control. Supplementation with aspartate had no effect on viability.

Because of the differences between the amino acids (i.e. lysine and threonine being protective and methionine enhancing killing), we have altered the conclusions of our revised submission.

Specifically, that in response to perturbed KatG activity, a metabolic rewiring reflects an attempt to maintain amino acid homeostasis whilst increasing or at least maintaining the production of constituents (i.e. lysine and threonine) that can provide protection against increases in oxidative stress.

These changes are in both the results and discussion of the revised manuscript. We have also added the figure below to the revised manuscript (Figure 4I and Extended Data Fig. 7k-m)

a) Extended Data Fig. 7k.

b) Extended Data Fig. 7l

c) Figure 4I

d) Extended Data Fig. 7m

Figure legend

The DS-parent or INH^R-katG were grown in stated media with 1 mM ascorbic acid and the stated amino acids at the following concentrations. (i.e. L-methionine (20 µg/ml), L-lysine (50 µg/ml), L-threonine (50µg/ml) and L-asparate (5 µg/ml). Amino acids were solubilised in dH₂O. Data is expressed as the reduction in viable colonies at each time point relative to the starting inoculum on day 0.

Reviewer #2 (Remarks to the Author):

Wang et al. use CRISPRi knockdowns to generate a map of vulnerabilities in an INH-resistant *katG* mutant of *Mtb*. They identify numerous vulnerabilities of a *katG* frameshift mutant. This is a very interesting approach that could serve as an example of how specific vulnerabilities of drug-resistant *Mtb* could be targeted for better therapy. The paper is broad rather than deep, but provides a mountain of potentially interesting data to mine.

Things for the authors to consider:

R2Q1: As the authors note in the Discussion, the clinical implication of their findings may be limited by their use of a *katG* frameshift mutant, when by far the most common INH-R *katG* mutant is S315T. The impact of the findings of this paper would be greatly strengthened if they could validate at least some of their findings in *katG*-S315T strains. Figure 5J is not convincing in this respect. The authors should test some of their hit genes in *katG*-S315T strains relative to the DS parent strain and see if the phenotypes observed in the *katG* frameshift mutant are also true in S315T.

In agreement with the reviewers comments we have performed additional experiments to demonstrate that clinically important vulnerabilities identified in INH^R-*katG* translate to clinically relevant INH^R genotypes.

First, using mycobacterial recombineering we have constructed clinically relevant genotypes that the World Health Organisation recognises as being associated with INH-resistance (Catalogue of mutations in *Mycobacterium tuberculosis* complex and their association with drug resistance, 2nd ed). As suggested by the reviewer we focused on KatG(S315T), in addition to an alternate KatGS315N mutation, the second most prevalent KatG mutation as reported by the WHO. Despite multiple attempts at recombineering, we were unable to generate a KatG(S315T) mutant. However, we were able to construct and confirm a KatGS315N-containing mutant. During the confirmation process, we also isolated a KatGS315N mutant that also contained a spontaneously evolved RIF-resistance RpoB mutation RpoBD435V. RpoBD435V is the second most prevalent RIF-resistance mutation in clinical populations after RpoBS450L. This gave us a INH^R-KatG^{S315N} and a Multi-drug resistant (MDR)-KatG^{S315N} mutant.

To test if the collateral vulnerabilities identified in our INH^R-*katG* mutant translated to clinical genotypes, we tested the sensitivity of INH^R-KatG^{S315N} and MDR-KatG^{S315N} to bedaquiline and linezolid. Both drugs are part of the new FDA approved BPAL regimen for the treatment of DR-*M. tuberculosis* and had increased efficacy against our INH^R-*katG* mutant. Neither the INH^R-KatG^{S315N} or MDR-KatG^{S315N} strains had increased sensitivity to killing by linezolid. However, comparable to INH^R-*katG*, both INH^R-KatG^{S315N} and MDR-KatG^{S315N} had increased sensitivity to both growth inhibition and killing by bedaquiline (Figure 6c-e and Extended Data Fig. 8h-i). This data is consistent with recent reports of bedaquiline hyper-susceptibility in INH^R-KatG^{S315T} clinical isolates of *M. tuberculosis* (<https://doi.org/10.1101/2023.10.17.562707>).

- a) Figure 6c
- b) Figure 6d
- c) Figure 6e
- d) Extended Data Fig. 8h
- e) Extended Data Fig. 8i

*All strains have pKM427 as this is an integrated plasmid used during the recombining process.

Second, we have also investigated whether the bedaquiline and linezolid hypersensitivity identified in our auxotrophic INH^R-katG mutant translated to virulent strains of *M. tuberculosis*. To do this we initially isolated INH^R strains in *M. tuberculosis* H37Rv using *in vitro* selection strategies. Using this approach we isolated two virulent mutants, INH^R-KatG^{M255R} and INH^R-KatG^{W351G}. Antimicrobial susceptibility testing demonstrated that INH^R-KatG^{M255R} had increased sensitivity to bedaquiline and linezolid, whilst INH^R-KatG^{W351G} had a comparable phenotype to the drug-susceptible parent strain (Figure below).

Further to this, and consistent with our INH^R-katG and INH^R-KatG^{S315N} mutant data, INH^R clinical isolates with a KatG^{S315T} mutation have increased susceptibility to bedaquiline (doi.org/10.1101/2023.10.17.562707). Also consistent with our data, INH^R-KatG^{S315T} clinical isolate had no increased susceptibility to linezolid (Figure below).

Figure 6a-b,f

This combined data demonstrates that collateral vulnerabilities identified in our INH^R-*katG* mutant can translate to clinically relevant genotypes and virulent strains of *M. tuberculosis*. Importantly, these combined findings suggest that the excellent efficacy of the BPaL regimen against DR-strains is the result of bedaquiline inadvertently targeting collateral vulnerabilities in DR strains of *M. tuberculosis*.

In light of our new data we have removed our analysis of the cryptic consortium data from our revised manuscript.

These changes are in both the results and discussion of the revised manuscript. We have also added the above figures to the revised manuscript (Figure 6, Extended Data Fig. 8h-l, and lines 346-378 in the revised manuscript).

R2Q2: Related to point 1 above, the paper would benefit from some small, confirmatory studies in proper, pathogenic Mtb. For example, some of the hits the authors follow up on focus on metabolic genes. Given that they work with a panleu auxotroph, it is unclear how this genetic background might impact some of the hits they follow up on.

Our use of *M. tuberculosis* strain mc²6206 in our work is a practical one. This strain can be used under PC2 conditions, and prevents potential risks of when working with drug-resistant strains that require PC3 containment.

We understand the reviewer's comments. Yet despite this, studies have demonstrated that when supplementing mc²6206 with pantothenic acid and leucine it behaves identically to virulent *M. tuberculosis* H37Rv (PMID: 31481950). It also retains a comparable rate of replication within macrophages and similar sensitivities to TB-agents (PMID: 31481950). Importantly, recent work examining the cell wall structure of various *M. tuberculosis* lab strains demonstrated that mc²6206 has a full complement of PDIM virulence lipids, a feature that is spontaneously lost from many virulent strains of *M. tuberculosis* due to routine lab culturing (PMID: 38740932). The loss of this PDIM layer in some PC3 strains can produce major discrepancies with clinical isolates.

To address the reviewer's concerns, we have constructed and assessed clinically relevant INH^R genotypes in *M. tuberculosis* strain mc²6206 and performed experiments using INH^R virulent strains of *M. tuberculosis*. See response to R2Q1.

R2Q3: The implicit assumption from the authors is that the level of knockdown at different ATc concentrations is the same between the DS and the INH-R strain. Given that this is a new genetic knockdown approach this should be validated.

We have performed additional experiments to test this. Specifically, we determined the level of transcriptional repression for four candidate genes (*gyrA*, *rpoB*, *mmpL3* and *qcrB*) at two ATc concentrations (i.e 30 and 300 ng/ml) in both the drug-susceptible parent and INH^R-*katG* strain. These results show very similar levels of transcriptional repression between both strains for all genes, particularly at the highest concentration of ATc (i.e. 300 ng/ml). Whilst there is some variation in transcriptional repression at 30 ng/ml ATc, both strains have a comparable level of transcriptional repression at the highest ATc concentration of 300 ng/ml.

This information and the figure below have been added to the revised manuscript as extended figure 1e. We have also added the following statement to the revised manuscript lines 104-106.

“At these concentrations of ATc, the level of transcriptional repression is comparable between both strains when tested by qPCR (Extended Data Fig. 1e-h)”

R2Q4: In Figure 1G, 388 hits genes seems like a lot (almost 10% of all Mtb genes). How many of these hit genes are essential in the DS strain? I would be curious how many essential genes come up as hits in their screen.

The number of genes that can be classified with altered essentiality in any given genetic context is dependent on both the severity of the parent phenotype and the interconnectivity of the mutated gene with other biological pathways. For example, *ctpC* is a component of mycobacterial oxidative stress response network and integrates a number of distinct biological pathways. Deletion of *ctpC* produces superoxide dismutase (SodA) hypomorph phenotype, and prior Tn-seq experiments in a *ctpC* mutant background identified 181 genes affecting the survival of this mutant (PMID: 26067605). This number of genes only reflects non-essential genes, and is in-line with our observations in INH^R-*katG*. Comparable results have been observed in *E. coli* Tn-seq screens when querying interactions between 163 core genes resulting in an average of 150 genetic interactions that aggravated the parent phenotype, again only reflecting the contribution from non-essential genes (PMID: 24586182). Furthermore, a recent WG-CRISPRi investigating more vulnerable genes in a *rpoB* rifampicin-resistant mutant of *M. tuberculosis* identified 99 genes of increased vulnerability. Whilst this value is lower than for our INH^R-*katG* mutant, our *katG* frameshift mutant is likely to have a more severe fitness cost than the RpoB(S450L) point mutant, and consequently be more susceptible to a wider range of perturbations. There are also differences in experimental design and analysis that likely contribute to differences in output. Of the 388 genes identified as more vulnerable in our screen, 316 are defined as being essential.

We have included a statement in our revised manuscript lines 149-150 that addresses this.

“This number of “more vulnerable” genes is in-line with other studies investigating interactions with genes that have core biological functions (33,34).”

R2Q5: The authors define genes as more vulnerable if they had 2 or more gRNAs that depleted more than 2 fold. How many genes had 2 or more gRNAs that enriched by more than 2 fold? How many genes had guides that both depleted and enriched?

Based on our selection criteria, only a small quantity of gRNAs (n=135) were enriched compared to the number of depleted gRNAs (n=1584). Of these enriched guides, only two genes (Rv0868c-moaD2 and Rv3375-amiD) had 2 or more guides in INH^R-*katG* that were more enriched by more than 2-fold, and neither of them had depleted guides. Given these criteria, none of the screened genes were both depleted and enriched.

Across both strains, 244 genes had at least 1 gRNA enriched and depleted. 125 were from the DS-parent, 127 were found in INH^R-*katG*, (8 genes were shared between them). All 251 genes with gRNAs that were both enriched and depleted only had 1 depleted gRNA, and up to 2 enriched gRNAs (3 genes had 2 enriched gRNA and everything else only had 1 enriched gRNA).

It is not uncommon to find gRNAs targeting the same genes behaving differently. This can be due to several factors, both technical and biological. For example, differences in gRNA binding affinity, secondary structure, or accessibility of the target site can influence binding efficacy (PMID: 34297925). Furthermore, for genes to be enriched they would have to induce a substantial increase in growth rate to be enriched. Yet due to the pooled nature of the screen it is harder to observed enriched genes than depleted as most of the population is unaffected by CRISPRi.

We agree with the reviewer this is an interesting topic, yet as only 2 genes were identified as enriched and unlikely to inform on future drug targeting we didn't investigated them further.

R2Q6: There seems to be no correlation between how the gene behaved in the screen and the fold reduction in the ATc MIC (Figure 2B). How do the authors interpret this?

We agree with the reviewer, that not all gRNAs identified in our screen as having increased efficacy against INH^R-*katG* had reduced ATc MICs relative to the DS-parent. We hypothesize that these differences are due to differences in experimental setup between our screen and ATc MICs, which relate to "target buffering". Some genes when targeted by CRISPRi have a buffering phenotype in which multiple cell divisions are needed to reduce a target protein below a critical threshold in order to observe a phenotype (PMID: 33238135). For this reason, some genes do not have a strong phenotype in ATc dose-response assays where the cultures are not back-diluted, whilst they have a stronger phenotype in the genetic screen where they are back-diluted (PMID: 33238135, 34346123).

In our original manuscript, this is best illustrated by *ideR*. Whilst the inhibition of *ideR* has a more inhibitory phenotype in INH^R-*katG* in ATc dose-response assays performed in 96 well plates (Figure 3f-see below), it does not reflect the large difference we initially observed in our genetic screen (Figure 3e-see below). However, we were able to reproduce the stronger vulnerability of *ideR* in INH^R-*katG* when we performed growth experiments that included a back-dilution every five days (Figure 3i-see below). Similar results were observed for *aroB* and *aroG* targeting gRNAs had no shift in ATc dose response assays in 96 well plates, but had increased activity against INH^R-*katG* when assessed in back-dilution growth assays (Extended Data Fig. 6h-i). Consequently, differences in experimental setup are the primary reason as to why there are differences between the level of vulnerability observed between our ATc MICs and genetic screens.

We have added the following to the results section of our revised manuscript lines 185-187.

“We hypothesised that differences in experimental setup were responsible for some gRNAs showing a poor correlation between the level of gRNA depletion in our screen and fold difference in ATc MIC (29)”.

R2Q7: The MIC experiments in Figure 2 and Ext Data Figure 3 are somewhat difficult to interpret with respect to the screen. In the screen, there are no hits at ATc-3 ng/ml. But in their follow-up, the MIC differences that the authors see between DS and INH-R tend to be at ATc concentrations below 10 ng/ml. For example, Figure 2C would suggest that *gyrB* hit at the ATc-3 screen, but perhaps even not a hit at higher ATc concentrations (ATc-30 and ATc-300). How do the authors explain this?

The reviewer is correct in that most gRNAs in our dose-response assays have an ATc MIC of around 10 ng/ml. This is in-line with our prior reports demonstrating that most essential genes have an ATc MIC between 3-30 ng/ml (PMID 34346123). As described for response for R2Q6 there are fundamental differences between the experimental set ups of our validation experiments in 96 well plates and those that are used for the WG-CRISPRi. These differences influence the efficacy of CRISPRi and can lead to differences in experimental outputs.

R2Q8: Related to point 6 above. A negative control gene ATc MIC experiment would be nice to include in the main text. Some of the differences between DS and INH-R are very subtle, so it would be good to know what a non-hit control gene looks like in this assay.

We agree with the reviewer that some of the differences in ATc MIC are subtle. Several hits from the screen did not validate in ATc MIC assays and had no difference in ATc MIC. These were included in the supplemental data Extended Data Fig. 6 (i.e. *aroG*(g6), *dapF*(g2), *aroB*(g6)) and were described in the text of the original manuscript.

R2Q9: The difference in the extent of kill between DS and INH-R is impressive (Fig 2H,I; Ext Fig 4). In many cases it seems as if the DS plateaus at a higher viable CFU than INH-R at higher drug concentrations. It would be good to also show data for MICs for these drugs. Is the difference the authors see due to decreased tolerance in INH-R? Decreased persistence? Something else?

We agree with the reviewer that in some instances the DS strain reaches a higher viable CFU at the end of the experiment. We conclude that this is largely experimental variation as in the majority of cases where this occurs the INH^R-katG started slightly lower, which continued to the final CFU count (i.e. 0.2-0.5 log₁₀ difference at the start is observed at the end). There is also variation in this trend as on occasion INH^R-katG starts lower and reaches the same final CFU (Extended Data Fig. 4a-c, e).

We agree with the reviewer with regards to the difference in killing between the DS-parent and *INH^R-katG*. We have included MIC data for each antibiotic included in our original study where we had investigated increased susceptibility to killing. In-line with the reviewers' comments, *INH^R-katG* has similar MIC to the majority of the tested antibiotics. We agree with the reviewer, that when this information is combined with the large differences in killing these results suggest the collateral vulnerabilities in *INH^R-katG* lead to a loss of antibiotic tolerance. Whilst this was mentioned in our original manuscript we have amended the revised manuscript to highlight this point. We have also added the MIC figures to the amended manuscript in extended data Fig 4 and 8.

Extended Data Fig. 4

Extended data Fig 8

R2Q10: The authors propose that INH-R remodels metabolism to compensate for perturbed KatG activity. This conclusion is quite interesting but is not clear that the authors demonstrate it to be true. That they see differences in metabolite levels is not necessarily demonstration that this is “compensation.” Compensation implies that there is a fitness benefit to such a change, which is not consistently borne out in their data.

We agree with the reviewer’s comments that our use of the term “compensation” may have been confusing. We have altered this in our revised submission and performed additional experiments to test the idea that altered metabolite levels could protect INH^R-katG from killing.

We performed time-kill experiments in both 7H9-OADC media and 7H12 with glucose as the sole carbon source. The primary difference between these media types is that 7H12 is composed of 7H9 salts with casamino acids, but does not contain the additional catalase that is present in OADC (C=catalase). We hypothesized that the absence of catalase would amplify the lethal effects of ascorbic acid. Consistent with this, ascorbic acid had increased lethality against both the DS_parent and INH^R-katG when grown in 7H12 media compared to 7H9 (Figure 4l and Extended Data Fig. 7k in the revised manuscript). These experiments showed that the addition of lysine or threonine was able to provide protection to INH^R-katG in both media conditions (Figure 4l and Extended Data Fig. 7k in the revised manuscript). Whilst the level of protection did not restore it to the levels of the DS-parent, it increased viability of INH^R-katG by >1 log₁₀ at multiple time points. The protection provided by exogenous lysine is consistent with depletion of *lysA* in the DS-parent having the greatest effect on viability against ascorbic acid compared to either *metA* or *aroK* depletion (Fig 4k of original submission). Interestingly, under both conditions the addition of exogenous methionine leads to increased

rates of killing, reaching the lower limit of detection early compared to non-supplemented media. Supplementation with aspartate had no effect on viability.

Because of the differences between the amino acids (i.e. lysine and threonine being protective and methionine enhancing killing) we have altered the conclusions of our revised submission. Specifically, that in response to perturbed KatG activity a rewiring of amino acid metabolism reflects an attempt to maintain amino acid homeostasis whilst increasing or at least maintaining the production of constituents (i.e. lysine and threonine) that can provide additional protection against increases in oxidative stress.

These changes are in both the results and discussion of the revised manuscript. We have also added the figure below to the revised manuscript (Figure 4l and Extended Data Fig. 7k-m)

- a) Extended Data Fig. 7k.
- b) Extended Data Fig. 7l
- c) Figure 4l
- d) Extended Data Fig. 7m

R2Q11: The connection with ribosome biogenesis is also quite interesting. Does knockdown of rplC or rpsB lead to ROS in the DS and INH-R strain? Do linezolid and kanamycin increase ROS levels in their strains?

Bactericidal antibiotics from diverse functional classes have been widely reported to increase ROS production and drive antibiotic lethality (PMID: 24803433, 30948634). Whilst we have not reported on whether the protein synthesis targeting transcriptional knockdowns (*rpIC* or *rpsB*) or antibiotics (linezolid or kanamycin) increase ROS, our original manuscript reported that the addition of redox stress generating compounds, i.e. H₂O₂, to which INH^R-*katG* has increased sensitivity leads to greater increases in ROS. Furthermore, a recent preprint (doi: <https://doi.org/10.1101/2023.10.17.562707>) has demonstrated that bedaquiline increases ROS in *M. tuberculosis katG* mutants. This is consistent with *M. tuberculosis katG* mutants having increased sensitivity to killing and inhibition by antibiotics, including BDQ, that increase the levels of ROS.

R2Q12: Figure 1G, it is not clear how these genes were called as hits. More explanation in the Figure Legend would be welcomed. This figure also doesn't convey what are the strongest hits from the screen. Is there an alternative way to display the screen to show both effect size and statistical significance, e.g. a volcano plot?

Additional explanations have been added to the legend of Figure 1g (lines 1227-1230) for clarification:

“Genes identified as being more vulnerable to inhibition in INH^R-katG are defined as when (i) they were called essential in INH^R-katG and (ii) had no less than 2 gRNAs that were depleted by >1-log₂ fold more than observed in the DS-parent.”

The criteria for genes being identified as hits were originally reported in the text and methods of our original manuscript. Differences in the magnitude of gRNA depletion between strains where originally presented in Figure 1f, and are available in the source data. Whether a gene is identified as being more vulnerable is a binary yes/no decision. Either a gene meets the criteria or it does not. Consequently, there is not a hierarchy of hit genes. For this reason, this data cannot be appropriately displayed in a volcano plot.

R2Q13: Figure 2A (and elsewhere) is labelled as “pathway enrichment analysis.” It is not clear from the figure how this analysis quantifies enrichment. Please explain. Is the enrichment statistically significant? What is the null hypothesis?

We agree with the reviewer that the term “enrichment” was confusing. We have removed this from the revised manuscript, and instead used the term pathway analysis. Figure 2a summarises the functional profiles of the identified “more vulnerable” genes in INH^R-*katG*. The x-axis quantifies the proportion of genes called “more vulnerable” over the total number of *M. tuberculosis* genes per functional pathway, and the y-axis showed the name of each functional pathway and the total number of genes in each class. Statistical significance was determined when identifying if a gRNA was depleted based on an exact test (Figure 1), data are available in SourceData_Fig1_a. Whether a gene is identified as being more vulnerable is a binary yes/no decision based on the number of significantly depleted guides and the differences in the magnitude of gRNA depletion between strains. The null hypothesis is there is no difference in gene vulnerabilities between INH^R-*katG* and its DS parent (mc²6206).

We have edited Figure legends in 2a, 4a, 5a, and Extended Data Fig. 2f for clarification

Reviewer #3 (Remarks to the Author):

Summary: Wang et al. report the generation of a whole genome CRISPRi library targeting *Mycobacterium tuberculosis* (Mtb) and its use to probe for collateral vulnerabilities that emerge in INH-resistant strains associated with mutations in the KatG catalase-peroxidase. Initial characterization of the CRISPRi library served to validate predictions of gene essentiality from a prior Mtb genome-wide CRISPRi screen reported by Bosch et al. By employing a multi-disciplinary approach consisting of CRISPRi screening in an INH-resistant katG mutant background couple with transcriptomics and metabolomics, they also identified multiply genes and pathways that exhibit enhanced vulnerability to inhibition (by transcriptional silencing) in a katG mutant vs. wild-type. These included genes involved in redox homeostasis, amino acid metabolism, and ribosome biogenesis. It is hypothesized that metabolic and transcriptional adaptation to compensate for the loss of KatG activity lead to “druggable vulnerabilities” that may offer novel strategies for treatment of drug-resistant Mtb. The potential for exploiting these vulnerabilities was highlighted by demonstration of improved potency of multiple classes of protein synthesis inhibitors against INHR katG mutants, both in vitro and in a macrophage infection model. This observation was further extended by analysis of pre-existing data on ~12,000 Mtb clinical isolates from the CRyPTIC database that revealed that a greater proportion of INHR strains had lower MICs for linezolid than INHS strains. Overall, this was a well-written and clearly organized manuscript with figures that effectively conveyed key aspects of a large dataset. Methods were, for the most part, described in sufficient detail and supplementary data was also very relevant. One significant weakness or concern relating to experimental design plus a few minor corrections/suggestions are detailed below.

Major:

R3Q1: The use of a katG frameshift mutant, rather than more clinically relevant S315T mutants (>70% of INH-R clinical isolates) introduces undesirable caveats into the interpretation of the data. While this is briefly acknowledged in the discussion (line 355), the implications of this choice should be elaborated on in more detail. S315T KatG has only mildly diminished catalase-peroxidase activity compared to katG null genotype expected in frameshift mutant. Given the modest level of impact of CRISPRi silencing of “hit” genes in the katG null mutant, the enhanced vulnerability of at least select target genes should be validated in the S315T katG background.

In agreement with reviewers comment we have performed additional experiments investigating if “hits” identified in INH^R-katG translate to clinically relevant INH^R KatG genotypes. These results highlight that clinically important vulnerabilities identified in INH^R-katG translate to clinically relevant INH^R genotypes.

First, using mycobacterial recombineering we have constructed clinically relevant genotypes that the World Health Organisation recognises as being clinically associated with INH-resistance (Catalogue of mutations in *Mycobacterium tuberculosis* complex and their association with drug resistance, 2nd ed). As suggested by the reviewer we focused on KatG(S315T), in addition to an alternate KatGS315N mutation, the second most prevalent KatG mutation as reported by the WHO. Whilst, despite multiple attempts, we were unable to generate a KatG(S315T) mutant we were able to construct and confirm a KatGS315N-containing mutant. During the confirmation process, we also isolated a KatGS315N mutant that also contained the RIF-resistance RpoB mutation RpoBD435V. RpoBD435V is the second most prevalent RIF-resistance mutation in clinical populations after RpoBS450L. This gave us an INH^R-KatG^{S315N} and a Multi-drug resistant (MDR)-KatG^{S315T} mutant.

To test if the collateral vulnerabilities identified in our INH^R-katG mutant translated to clinical genotypes, we tested the sensitivity of INH^R-KatG^{S315N} and MDR-KatG^{S315N} to bedaquiline and linezolid. Both drugs are part of the new FDA approved BPAL regimen for the treatment of DR-*M. tuberculosis* and had increased efficacy against our INH^R-katG mutant. Neither the INH^R-

KatG^{S315N} or MDR-KatG^{S315N} strains had increased sensitivity to killing by linezolid. However, comparable to INH^R-katG, both INH^R-KatG^{S315N} and MDR-KatG^{S315N} had increased sensitivity to both growth inhibition and killing by bedaquiline (Figure 6c-e and Extended Data Fig. 8h-i). This data is consistent with recent reports of bedaquiline hyper-susceptibility in INH^R-KatG^{S315T} clinical isolates of *M. tuberculosis* (<https://doi.org/10.1101/2023.10.17.562707>).

In the figure below, all strains have pKM427 as this is an integrated plasmid used during the recombineering process.

- a) Figure 6c
- b) Figure 6d
- c) Figure 6e
- d) Extended Data Fig. 8h
- e) Extended Data Fig. 8i

Second, we have also investigated whether the bedaquiline and linezolid hypersensitivity identified in our auxotrophic INH^R-katG mutant translated to virulent strains of *M. tuberculosis*. To do this we initially isolated INH^R strains in *M. tuberculosis* H37Rv using *in vitro* selection strategies. Using this approach we isolated two virulent mutants, INH^R-KatG^{M255R} and INH^R-KatG^{W351G}. Antimicrobial susceptibility testing demonstrated that INH^R-KatG^{M255R} had increased sensitivity to bedaquiline and linezolid, whilst INH^R-KatG^{W351G} had a comparable phenotype to the drug-susceptible parent strain (Figure below).

Further to this, and consistent with our INH^R-*katG* and INH^R-KatG^{S315N} mutant data, INH^R clinical isolates with a KatG^{S315T} mutation have increased susceptibility to bedaquiline (doi.org/10.1101/2023.10.17.562707). Also consistent with our data, INH^R-KatG^{S315T} clinical isolates had no increased susceptibility to linezolid (Figure below).

Figure 6a-b,f

This combined data demonstrate that collateral vulnerabilities identified in our INH^R-*katG* mutant can translate to clinically relevant genotypes and virulent strains of *M. tuberculosis*. Importantly, these combined findings suggest that the excellent efficacy of the BPaL regimen against DR-strains is the result of bedaquiline inadvertently targeting collateral vulnerabilities in DR strains of *M. tuberculosis*.

We also agree with the reviewer, that our original analysis of data from the cryptic consortium was not complete, and in light of our new data have removed this from our revised manuscript.

These changes are in both the results and discussion of the revised manuscript. We have also added the above figures to the revised manuscript (Figure 6, Extended Data Fig. 8h-l, and lines 346-378 in the revised manuscript).

R3Q2: Statistical analysis of quantitative phenotypic data (e.g. MICs) were not included. For example, the authors claimed that when gene *isc* was targeted there was impairment in intracellular survival (line 167-68, Extended Data Fig. 3n). However, the figure shows that both DS-Parent and INH^R-*katG* had survival above 100% and only a marginal difference is evident. Was any statistical analysis done to validate the difference?

We agree with the reviewer, and have removed the marginal difference in intracellular survival when expressing the *iscS* gRNA from the manuscript text.

Minor:

R3Q3: Line 40 – Suggest updating reference to six-month regimen to recently approved 4 month RPT-MOX regimen for drug-susceptible TB.

This has been amended in the revised manuscript line 50

R3Q4: Line 50-51 – Either here or in discussion, it would be helpful to elaborate on the types and frequency of clinically relevant mutations in *katG* and their different impacts on catalase-peroxidase activity, fitness, and virulence phenotypes (see major concern above).

We have added additional discussion addressing this point. This can be found in both the introduction and discussion of our revised manuscript.

Added discussion in lines 434-450 of the revised manuscript:

“The majority of clinical INHR^R strains harbour a substitution at KatG Serine 315 to either threonine, asparagine or arginine. The selection against loss of function mutants, is likely due to a need to preserve at least partial KatG catalase/oxidase activity to survive within a host-environment (76). By constructing a INHR and MDR strains that contained a KatGS315N mutation, instead of the katG frameshift mutant used in our WG-CRISPRi screen, we were able to confirm that our identified collateral vulnerabilities could translate to clinically relevant genotypes. Importantly, the increased sensitivity of both INHR-KatGS315N and MDR-KatGS315N to bedaquiline is consistent with recent reports of bedaquiline hyper-susceptibility in INHR KatGS315T clinical isolates of M. tuberculosis (46). Interestingly neither the INHR-KatGS315N or MDR-KatGS315N mutant had the increased sensitivity to linezolid that was observed for our katG frameshift mutant. These combined findings suggest that the excellent efficacy of the BPaL regimen may be a result of bedaquiline inadvertently targeting collateral vulnerabilities in DR strains of M. tuberculosis.”

R3Q5: Line 103-104 – Do not think that “disagreed essential genes” is grammatically correct. Perhaps “discrepant” or “genes with discrepant essentiality predictions”

Agreed, terms have been changed to “genes with discrepant essentiality predictions” in the revised manuscript lines 122-125.

R3Q6: Line 125 – Should this read “depleted by >1 log₂ fold change MORE than observed in the DS-parent”?

This has been updated in the revised manuscript line 145.

R3Q7: Line 218 – The statement that “DosR regulates survival under oxidative stress” does not seem to accurately reflect the role of DosR under hypoxic stress (basically the opposite). This would typically mean that for e.g. a dosR- mutant would be hypersusceptible to H₂O₂ or other oxidants, and to my knowledge that is not the case. Please clarify this statement.

Prior work has demonstrated that DosR regulates the expression of genes in response to both redox and hypoxic stress. DosS and DosT are both sensor kinases that interact with and activate DosR in response to distinct modulatory ligands. Specifically DosS functions as a redox signal, whilst DosT functions as a hypoxia sensor (PMID: 17609369).

We have amended the revised manuscript lines 249-255 to clarify our original statement and line of investigation.

“In line with a recent transcriptional study of a M. tuberculosis katG mutant, we observed an upregulation of DosR and 33 genes (i.e. 65%) in the DosR regulon in INHR-katG (Table S3) (46). Interactions with the redox sensors DosS and DosT allow DosR to regulate genes in response to both redox stress and the transition to hypoxia (47–52). Based on this, we hypothesised that in addition to increased sensitivity to redox stress, INHR-katG would have an altered ability to survive a hypoxic environment.”

R3Q8: Line 372-373 – The double use of translation in “The translation of this genetic data translated to existing chemical inhibitors” is unnecessary.

We agree with the reviewer and have amended the second use of this statement in the revised manuscript lines 456-457.

R3Q9: Line 375-376 - What is a “portable CRISPRi platform”? Given the species-specificity of promoters, it seems unlikely that a single platform would work across diverse pathogens.

The term "portable CRISPRi platform" refers to the adaptability and versatility of employing the CRISPR interference (CRISPRi) system to identify genetic vulnerabilities across diverse bacterial species. While it is true that our current promoters are designed specifically for *Mycobacterium tuberculosis*, CRISPRi technology has been successfully applied across eight bacterial phyla (PMID: 35813973).

Some pathogens have genome-scale CRISPRi libraries available, which could be adapted for gene vulnerabilities, such as *E. coli* (PMID: 29946130) and *Streptococcus pneumoniae* (PMID: 33120116).

For pathogens without available promoters and libraries, resources are available for promoter and library development. For example, addgene (<https://www.addgene.org/>) offers a vast array of CRISPR-related plasmids and can facilitate the development of custom promoters and libraries tailored to specific taxa.

To avoid confusion around, we have removed this statement from the end of the discussion in the revised manuscript.

R3Q10: Line 394: Please clarify what is meant by “functional 1.5% agar”.

The word “functional” had been deleted in the revised manuscript line 475

R3Q11: Maintain consistency with either number or word usage at the beginning of a sentence (100µl in Line 640 and One hundred µl in Line 671).

We have changed 100 µl to One hundred µl in the revised manuscript line 733

R3Q12: Line 765-66: Please provide a link to scripts in GitHub.

Link added in the revised manuscript line 857

R3Q13: The number of replicates of each experiment, particularly the phenotypic characterization of single-gene CRISPRi knockdowns, could not be determined. Please clarify the number of technical and biological replicates for each assay type.

For the high-throughput CRISPRi screen of each strain, 5 replicates were performed per ATc concentration. This was described in our original manuscript. Information related to the numbers of replications (both biological and technical) for all other experiments (e.g. single gene CRISPRi knockdown, drug-susceptibility etc) have been added to the legends of relevant figures.

R3Q14: Line 1097 – Check grammar “genes that are of more vulnerable” should read “genes that are more vulnerable”.

The sentence has been corrected as suggested in the revised manuscript line 1237.

R3Q15: Figure 3 legend (lines 1148-1153) – There is a panel K but no description of (k) in the legend. Also, the legend for (i) and (j) seems to correspond to panels J and K, respectively. Also, (f) and (g) leading into description of these panels are not bolded, whereas letters for other panel descriptions are.

Agreed, changes have been made as suggested. New legend descriptions for panel (i) has been added in the revised manuscript lines 1291-1294:

“(i) Growth kinetics of M. tuberculosis DS-parent and INHR-katG expressing gRNA targeting ideR, Bacterial growth was measured by OD600. All strains were grown in 7H9 media with ATc-300 and back diluted 1/20 with fresh media on days 5 and 10. Data is mean \pm SD of three biological replicates, (n=2).”

R3Q16: Line 1185 – Should be “at the stated concentration”. Also, should (m) and (n) be bolded?

Typo has been corrected as suggested in the revised manuscript line 1340

R3Q17: Providing additional details such as MIC values about certain experiments would be helpful. Specifically, in Figure 5 Linezolid was used as 1X-9X MIC in macrophage survival assay without mention of MIC value in graphs or in text section. This info could either be integrated into the graphs or as a supplementary table or in some other way.

We have added this information for Figure 5 and the new Figure 6 to the legends in the revised manuscript.

R3Q18: Convert MIC values in either $\mu\text{g/ml}$ or μM for consistency and quick comparison of LZD sensitivity between INHR-katG and clinical isolates. LZD in $\mu\text{g/ml}$ was used in assays with clinical isolates whereas in other assays it was in μM concentration (Fig.5).

We have removed the $\mu\text{g/ml}$ from the manuscript, and have used μM throughout.

R3Q19: Maintain consistency in italicizing scientific names in reference section.

We have updated the references for consistency.

R3Q20: Should in-text citations be as underlined superscript numbers as per journal recommendation e.g. 34 instead of (34)? Maybe this is done in proof stage.

To the best of our knowledge, the original submission did not require references to be superscript.

R3Q21: Could not find the RNA sequencing data using provided accession number (PRJNA1041353). If this bioproject is locked until time of publication, can disregard.

Yes, the raw data of this project is locked until the time of publication, sequencing data can be viewed with this temporary reviewer link: (<https://dataview.ncbi.nlm.nih.gov/object/PRJNA1041353?reviewer=h8q090us3ljgsnafo4co1e3hkr>)

- Spell checks needed.

R3Q22: Line 468: (Quibt)

Changed to “Qubit” in line 554

R3Q23: Line 403: (pJLR965)

Changed to “pLJR965” in revised manuscript line 487

R3Q24: Line 610 (CRISRPi)

Changed to “CRISPRi” in line 703

R3Q25: Line 750 (fremoved)

Changed to “removed” in line 841

R3Q26: Remove an extra period (Line 1146) and add a period (Line 634).

Corrections have been made as suggested.

R3Q27: Maintain consistency with either number or word usage at the beginning of a sentence (100µl in Line 640 and One hundred µl in 671).

We have changed 100 µl to One hundred µl in the revised manuscript line 733

Reviewer #4 (Remarks to the Author):

REVIEWERS' COMMENTS

Reviewer #1 (Remarks to the Author):

The authors have address the comments convincingly by adding the relevant supplementary data and amending the main text accordingly.

Reviewer #2 (Remarks to the Author):

The authors have done a commendable job in addressing my comments. I have only two remaining points I think the authors should address.

R2Q1: It is unfortunate that the authors were not able to generate a katG-S315T mutant by recombineering. S315T represents >94% of all INH-R Mtb in the clinic (PMID: 26104204). katG-S315N is very rare (1-2% of INH-R?). So, confirmation of some of the authors phenotypes in S315N does not broaden the impact of the authors conclusions the way S315T would have. The authors cite a preprint from the Yang lab which shows increased sensitivity of INH-R clinical strains (relative to INH-S clinical strains) at a single dose of BDQ, an experiment that includes two katG-S315T strains. While this experiment is consistent with the authors conclusion, it is not definitive as presented. Barring further experiments to address this point, I would recommend that the authors be explicit in their conclusions that further work is necessary to definitively show whether the vulnerabilities they identify extend beyond catalase-dead INH-R Mtb strains to INH-R strains like katG-S315T that retain near wild-type levels of catalase activity.

R2Q4: How many genes are essential by their CRISPRi method? From prior Tn-seq experiments in Mtb, the number is ~650. This would suggest that at their highest dose of ATc, more than half of all essential genes are now more essential in their INH-R strain. This seems biologically unlikely, and I would tend to believe their analytic method is over-calling hit genes. I would recommend the authors at a minimum modify Figure 1 to make it clear how many hit genes are essential vs non-essential, and what fraction of each gene category are being called hit genes.

Reviewer #3 (Remarks to the Author):

I have conferred with the Early Career Researcher who co-reviewed this manuscript with me, and we both concur that the authors have satisfactorily addressed our concerns. The investigators have conducted a considerable amount of additional experiments to provide ket data requested by us as well as other reviewers. They have provided in depth justifications for their interpretations of data or choices of experiments to conduct and clarified any points of confusion. As a result of their rigorous efforts to address all critiques, the revised manuscript has been improved and is suitable for publication.

Reviewer #3 (Remarks on code availability):

Not qualified to review the code.

Reviewer #4 (Remarks to the Author):

Reviewer comments in black
Our responses are in blue.

REVIEWER COMMENTS

Reviewer #2 (Remarks to the Author)

The authors have done a commendable job in addressing my comments. I have only two remaining points I think the authors should address.

R2Q1: It is unfortunate that the authors were not able to generate a *katG*-S315T mutant by recombineering. S315T represents >94% of all INH-R Mtb in the clinic (PMID: 26104204). *katG*-S315N is very rare (1-2% of INH-R?). So, confirmation of some of the authors phenotypes in S315N does not broaden the impact of the authors conclusions the way S315T would have. The authors cite a preprint from the Yang lab which shows increased sensitivity of INH-R clinical strains (relative to INH-S clinical strains) at a single dose of BDQ, an experiment that includes two *katG*-S315T strains. While this experiment is consistent with the authors conclusion, it is not definitive as presented. Barring further experiments to address this point, I would recommend that the authors be explicit in their conclusions that further work is necessary to definitively show whether the vulnerabilities they identify extend beyond catalase-dead INH-R Mtb strains to INH-R strains like *katG*-S315T that retain near wild-type levels of catalase activity.

We agree with the reviewer that it was unfortunate that we were unable to generate a INH^R-KatG^{S315T} strain. The KatG^{S315T} mutation is the dominate clinical genotype, although it represents 77.8% of INH-resistant clinical isolates rather than the >94% suggested by the reviewer (*WHO, 2023, Catalogue of mutations in Mycobacterium tuberculosis complex and their association with drug resistance*). Whilst the INH^R-*katG*^{S315N} is less frequent, found in 1-2% of INH-resistant isolates, the mutation has been graded as being associated with INH-resistance (*WHO, 2023, Catalogue of mutations*). Thus our experiments with the INH^R-*katG*^{S315N} demonstrate that collateral vulnerabilities identified in INH^R-*katG* translate to clinically relevant INH^R genotypes. We have expanded our introduction and discussion to address the reviewers concerns.

We have modified the introduction to highlight the frequency of different mutations:

61-64: "Whilst a large variety of *katG* mutations have been reported, the majority of clinical INH^R strains harbour a substitution at KatG Serine 315 to threonine (77.8%), with mutations to asparagine (1%) or arginine (0.1%) being observed at a lower frequency (11)."

And added the sentence

380-381: "Future research is essential to validate these vulnerabilities in alternative clinically relevant INH^R strains."

R2Q4: How many genes are essential by their CRISPRi method? From prior Tn-seq experiments in Mtb, the number is ~650. This would suggest that at their highest dose of ATc, more than half of all essential genes are now more essential in their INH-R strain. This seems biologically unlikely, and I would tend to believe their analytic method is over-calling hit genes. I would recommend the authors at a minimum modify Figure 1 to make it clear how many hit genes are essential vs non-essential, and what fraction of each gene category are being called hit genes.

Based on our screen that targeted 3991 protein-encoding genes, 631 were called essential and 3360 were non-essential in the INH^R-*katG* strain. A total of 16.3% of genes screened in the study were called essential, with approximately 9.7% (n =388) were called more essential in the INH^R-*katG* strain. We have modified the Figure legend to include the numbers listed above to clarify.

Further to this, the number of genes classified with altered essentiality is dependent on the genetic context. A more severe parent phenotype is more likely to have interactions with a greater number of other biological pathways. For example, *ctpC* a component of mycobacterial oxidative stress response network, interacted with 181 genes from a prior Tn-seq experiments that only investigated non-essential genes (PMID: 26067605). Comparable results have been observed in *E. coli* Tn-seq screens where core genes had on average of 150 genetic interactions with non-essential genes (PMID: 24586182). Furthermore, a recent WG-CRISPRi investigating more vulnerable genes in a *rpoB* rifampicin-resistant mutant of *M. tuberculosis* identified 99 genes of increased vulnerability. Whilst this value is lower than

for our INH^R-*katG* mutant, our *katG* frameshift mutant as a more severe fitness cost than the RpoB(S450L) point mutant, and consequently be more susceptible to a wider range of perturbations.

We have included a statement in our revised manuscript lines 218-219 that addresses this.

“This number of “more vulnerable” genes is in-line with other studies investigating interactions with genes that have core biological functions (33,34).”